# DiffusionPDE: Generative PDE-Solving Under Partial Observation

**Jiahe Huang**[1]    **Guandao Yang**[2]    **Zichen Wang**[1]    **Jeong Joon Park**[1]

[1]University of Michigan
[2]Stanford University
{chloehjh, zzzichen, jjparkcv}@umich.edu
guandao@stanford.edu

## Abstract

We introduce a general framework for solving partial differential equations (PDEs) using generative diffusion models. In particular, we focus on the scenarios where we do not have the full knowledge of the scene necessary to apply classical solvers. Most existing forward or inverse PDE approaches perform poorly when the observations on the data or the underlying coefficients are incomplete, which is a common assumption for real-world measurements. In this work, we propose *DiffusionPDE* that can simultaneously fill in the missing information and solve a PDE by modeling the joint distribution of the solution and coefficient spaces. We show that the learned generative priors lead to a versatile framework for accurately solving a wide range of PDEs under partial observation, significantly outperforming the state-of-the-art methods for both forward and inverse directions. See our project page for results: jhhuangchloe.github.io/Diffusion-PDE/.

## 1 Introduction

Partial differential equations (PDEs) are a cornerstone of modern science, underpinning many contemporary physical theories that explain natural phenomena. The ability to solve PDEs grants us the power to predict future states of a system (forward process) and estimate underlying physical properties from state measurements (inverse process).

To date, numerous methods [1, 2] have been proposed to numerically solve PDEs for both the forward and inverse directions. However, the classical methods can be prohibitively slow, prompting the development of data-driven, learning-based solvers that are significantly faster and capable of handling a family of PDEs. These learning-based approaches [3–6] typically learn a *deterministic* mapping between input coefficients and their solutions using deep neural networks.

Despite the progress, existing learning-based approaches, much like classical solvers, rely on complete observations of the coefficients to map solutions. However, complete information on the underlying physical properties or the state of a system is rarely accessible; in reality, most measurements are sparse in space and time. Both classical solvers and the state-of-the-art data-driven models often overlook these scenarios and consequently fail when confronted with partial observations. This limitation confines their use primarily to synthetic simulations, where full scene configurations are available by design, making their application to real-world cases challenging.

We present a comprehensive framework, DiffusionPDE, for solving PDEs in both forward and inverse directions under conditions of highly partial observations—typically just 1~3% of the total information. This task is particularly challenging due to the numerous possible ways to complete missing data and find subsequent solutions. Our approach uses a generative model to formulate the joint distribution of the coefficient and solution spaces, effectively managing the uncertainty and simultaneously reconstructing both spaces. During inference, we sample random noise and iteratively

38th Conference on Neural Information Processing Systems (NeurIPS 2024).

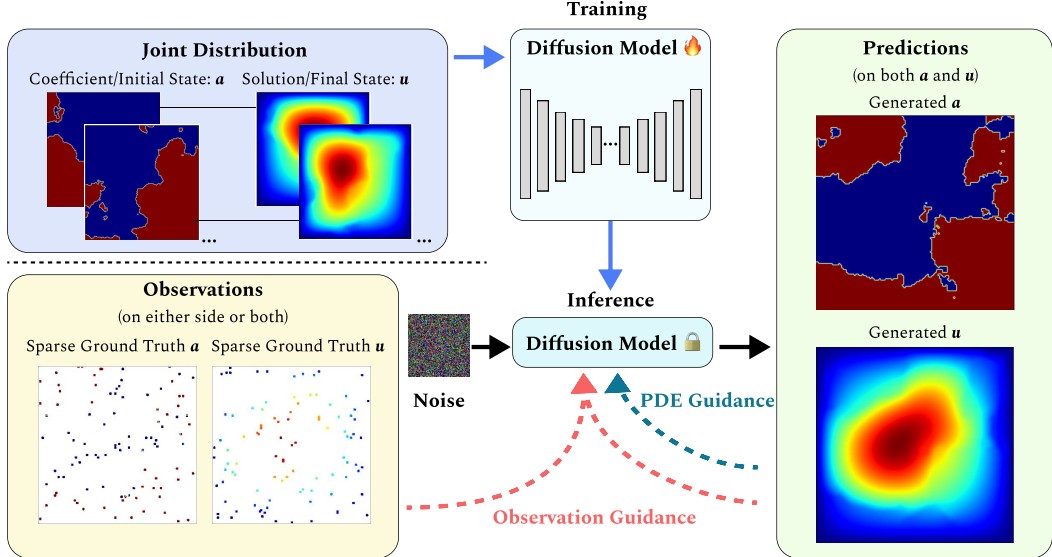

Figure 1: We propose DiffusionPDE, a generative PDE solver under partial observations. Given a family of PDE with coefficient (initial state) $a$ and solution (final state) $u$, we train the diffusion model on the joint distribution of $a$ and $u$. During inference, we gradually denoise a Gaussian noise, guided by sparse observation and known PDE function, to recover the full prediction of both $a$ and $u$ that align well with the sparse observations and the given equation.

denoise it following standard diffusion models [7]. However, we uniquely guide this denoising process with sparse observations and relevant PDE constraints, generating plausible outputs that adhere to the imposed constraints. Notably, DiffusionPDE can handle observations with arbitrary density and patterns with a single pre-trained generative network.

We conduct extensive experiments to show the versatility of DiffusionPDE as a general PDE-solving framework. We evaluate it on a diverse set of static and temporal PDEs, including Darcy Flow, Poisson, Helmholtz, Burger's, and Navier-Stokes equations. DiffusionPDE significantly outperforms existing state-of-the-art learning-based methods for solving PDEs [3–6, 8] in both forward and inverse directions with sparse measurements, while achieving comparable results with full observations. Highlighting the effectiveness of our model, DiffusionPDE accurately reconstructs the complete state of Burgers' equation using time-series data from just five sensors (Fig. 4), suggesting the potential of generative models to revolutionize physical modeling in real-world applications.

## 2   Related Works

Our work builds on the extensive literature of three areas: forward PDE solvers, inverse PDE solvers, and diffusion models. Please see relevant surveys for more information [9–13].

**Forward PDE Solvers.**   PDE solvers take the specification of a physics system and predict its state in unseen space and time by solving an equation involving partial derivatives. Since Most PDEs are very challenging to solve analytically, people resolve to numerical techniques, such as Finite Element Method [14, 2] and Boundary Element Method [1, 15]. While these techniques show strong performance and versatility in some problems, they can be computationally expensive or difficult to set up for complex physics systems. Recently, advancements in deep-learning methods have inspired a new set of PDE solvers. Raissi et al. [16, 6] introduce Physics-Informed Neural Networks (PINNs), which optimize a neural network using PDE constraints as self-supervised losses to output the PDE solutions. PINNs have been extended to solving specific fluid [17, 18], Reynolds-averaged Navier–Stokes equations [19], heat equations [20], and dynamic power systems [21]. While PINNs can tackle a wide range of complex PDE problems, they are difficult to scale due to the need for network optimization. An alternative approach, neural operators [3, 5], directly learn the mapping from PDE parameters (*e.g.*initial and boundary condition) to the solution function. Once trained, this method avoids expensive network optimization and can instantly output the solution result. This idea

has been extended to solve PDE in 3D [22, 23] , multiphase flow [24], seismic wave [25, 26], 3D turbulence [27, 28], and spherical dynamics [29]. People have also explored using neural networks as part of the PDE solver, such as compressing the physics state [30–33]. These solvers usually assume known PDE parameters, and applying them to solve the inverse problem can be challenging.

**PDE inverse problem.** The inverse problem refers to finding the coefficients of a PDE that can induce certain observations, mapping from the solution of a PDE solver to its input parameters. People have tried to extend traditional numerical methods to this inverse problem [34–38], but these extensions are non-trivial to implement efficiently. There are similar attempts to inverse deep-learning PDE solvers. For example, one can inverse PINNs by optimizing the network parameters such that their outputs satisfy both the observed data and the governing equations. iFNO [39] and NIO [40] tries to extend FNO [3]. Other methods [41, 42] directly learn the operator functions for the inverse problem. PINO [4] further combines neural operators with physics constraints to improve the performance of both forward and inverse problems. These methods assume full observations are available. To address the inverse problem with partial observations, people have tried to leverage generative priors with Graph neural networks [43, 8]. These works have not demonstrated the ability to solve high-resolution PDEs, possibly limited by the power of generative prior. We want to leverage the state-of-the-art generative model, diffusion models, to develop a better inverse PDE solver.

**Diffusion models.** Diffusion models have shown great promise in learning the prior with higher resolutions by progressively estimating and removing noise. Models like DDIM [44], DDPM [7], and EDM [45] offer expressive generative capabilities but face challenges when sampling with specific constraints. Guided diffusion models [46–49] enhance generation processes with constraints such as image inpainting, providing more stable and accurate solutions. Prior works on diffusion models for PDEs highlight the potential of diffusion approaches by generating PDE datasets such as 3D turbulence [50, 51] and Navier-Stokes equations [52] with diffusion models. Diffusion models can also be used to model frequency spectrum and denoise the solution space [53], and conditional diffusion models are applied to solve 2D flows with sparse observation [54]. However, the application of diffusion models to solve inverse problems under partial observation remains underexplored. In this work, we aim to take the initial steps towards addressing this gap.

## 3  Methods

### 3.1  Overview

To solve physics-informed forward and inverse problems under uncertainty, we start by pre-training a diffusion generative model on a family of partial differential equations (PDEs). This model is designed to learn the joint distribution of the PDE coefficients (or the initial state) and its corresponding solutions (or the final state). Our approach involves recovering full data in both spaces using sparse observations from either or both sides. We achieve this through the iterative denoising of random Gaussian noise as in regular diffusion models but with additional guidance from the sparse observations and the PDE function enforced during denoising. The schematic description of our approach is shown in Fig. 1.

### 3.2  Prelimary: Diffusion Models and Guided Diffusion

Diffusion models involve a predefined forward process that gradually adds Gaussian noise to the data and a learned reverse process that denoises the data to reconstruct the original distribution. Specifically, Song et al. [55] propose a deterministic diffusion model that learns an $N$-step denoising process that eventually outputs a denoised data $\boldsymbol{x}_N$ and satisfies the following ordinary differential equations (ODE) at each timestep $t_i$ where $i \in \{0, 1, ..., N-1\}$

$$\mathrm{d}\boldsymbol{x} = -\dot{\sigma}(t)\sigma(t)\nabla_{\boldsymbol{x}} \log p\big(\boldsymbol{x}; \sigma(t)\big)\mathrm{d}t. \tag{1}$$

Here $\nabla_{\boldsymbol{x}} \log p\big(\boldsymbol{x}; \sigma(t)\big)$ is the score function [56] that helps to transform samples from a normal distribution $\mathcal{N}(0, \sigma(t_0)^2\mathbf{I})$ to a target probability distribution $p(\boldsymbol{x}; \sigma(t))$. To estimate the score function, Karras et al. [45] propose to learn a denoiser function $D(\boldsymbol{x}; \sigma)$ such that

$$\nabla_{\boldsymbol{x}} \log p\big(\boldsymbol{x}; \sigma(t)\big) = (D(\boldsymbol{x}; \sigma(t)) - \boldsymbol{x})/\sigma(t)^2 \tag{2}$$

To enable control over the generated data, guided diffusion methods [48] add guidance gradients to the score function during the denoising process. Recently, diffusion posterior sampling (DPS) [46]

**Algorithm 1** Sparse Observation and PDE Guided Diffusion Sampling Algorithm.

---

1: **input** DeterministicSampler $D_\theta(\boldsymbol{x}; \sigma)$, $\sigma(t_{i \in \{0,\dots,N\}})$, TotalPointCount $m$, ObservedPointCount $n$, Observation $\boldsymbol{y}$, PDEFunction $f$, Weights $\zeta_{obs}, \zeta_{pde}$

2: **sample** $\boldsymbol{x}_0 \sim \mathcal{N}(\mathbf{0}, \sigma(t_0)^2 \mathbf{I})$          ▷ Generate initial sampling noise

3: **for** $i \in \{0, \dots, N-1\}$ **do**

4:     $\hat{\boldsymbol{x}}_N^i \leftarrow D_\theta(\boldsymbol{x}_i; \sigma(t_i))$          ▷ Estimate the denoised data at step $t_i$

5:     $\boldsymbol{d}_i \leftarrow (\boldsymbol{x}_i - \hat{\boldsymbol{x}}_N^i)/\sigma(t_i)$          ▷ Evaluate $\mathrm{d}\boldsymbol{x}/\mathrm{d}\sigma(t)$ at step $t_i$

6:     $\boldsymbol{x}_{i+1} \leftarrow \boldsymbol{x}_i + (\sigma(t_{i+1}) - \sigma(t_i))\boldsymbol{d}_i$          ▷ Take an Euler step from $\sigma(t_i)$ to $\sigma(t_{i+1})$

7:     **if** $\sigma(t_{i+1}) \neq 0$ **then**

8:        $\hat{\boldsymbol{x}}_N^i \leftarrow D_\theta(\boldsymbol{x}_{i+1}; \sigma(t_{i+1}))$          ▷ Apply 2$^{\text{nd}}$ order correlation unless $\sigma = 0$

9:        $\boldsymbol{d}_i' \leftarrow (\boldsymbol{x}_{i+1} - \hat{\boldsymbol{x}}_N^i)/\sigma(t_{i+1})$          ▷ Evaluate $\mathrm{d}\boldsymbol{x}/\mathrm{d}\sigma(t)$ at step $t_{i+1}$

10:        $\boldsymbol{x}_{i+1} \leftarrow \boldsymbol{x}_i + (\sigma(t_{i+1}) - \sigma(t_i))\left(\frac{1}{2}\boldsymbol{d}_i + \frac{1}{2}\boldsymbol{d}_i'\right)$    ▷ Apply the trapezoidal rule at step $t_{i+1}$

11:     **end if**

12:     $\mathcal{L}_{obs} \leftarrow \frac{1}{n}\|\boldsymbol{y} - \hat{\boldsymbol{x}}_N^i\|_2^2$          ▷ Evaluate the observation loss of $\hat{\boldsymbol{x}}_N^i$

13:     $\mathcal{L}_{pde} \leftarrow \frac{1}{m}\|\mathbf{0} - f(\hat{\boldsymbol{x}}_N^i)\|_2^2$          ▷ Evaluate the PDE loss of $\hat{\boldsymbol{x}}_N^i$

14:     $\boldsymbol{x}_{i+1} \leftarrow \boldsymbol{x}_{i+1} - \zeta_{obs}\nabla_{\boldsymbol{x}_i}\mathcal{L}_{obs} - \zeta_{pde}\nabla_{\boldsymbol{x}_i}\mathcal{L}_{pde}$    ▷ Guide the sampling with $\mathcal{L}_{obs}$ and $\mathcal{L}_{pde}$

15: **end for**

16: **return** $\boldsymbol{x}_N$          ▷ Return the denoised data

---

made notable progress in guided diffusion for tackling various inverse problems. DPS uses corrupted measurements $\boldsymbol{y}$ derived from $\boldsymbol{x}$ to guide the diffusion model in outputting the posterior distribution $p(\boldsymbol{x}|\boldsymbol{y})$. A prime application of DPS is the inpainting problem, which involves recovering a complete image from sparsely observed pixels, which suits well with our task. This approach modifies Eq. 1 to

$$\mathrm{d}\boldsymbol{x} = -\dot{\sigma}(t)\sigma(t)\big(\nabla_{\boldsymbol{x}}\log p(\boldsymbol{x};\sigma(t)) + \nabla_{\boldsymbol{x}}\log p(\boldsymbol{y}|\boldsymbol{x};\sigma(t))\big)\mathrm{d}t. \tag{3}$$

DPS [46] showed that under Gaussian noise assumption of the sparse measurement operator $\mathcal{M}(\cdot)$, i.e., $\boldsymbol{y}|\boldsymbol{x} \sim \mathcal{N}(\mathcal{M}(\boldsymbol{x}), \delta^2\mathbf{I})$ with some S.D. $\delta$, the log-likelihood function can be approximated with:

$$\nabla_{\boldsymbol{x}}\log p(\boldsymbol{y}|\boldsymbol{x}_i;\sigma(t_i)) \approx \nabla_{\boldsymbol{x}_i}\log p(\boldsymbol{y}|\hat{\boldsymbol{x}}_N^i;\sigma(t_i)) \approx -\frac{1}{\delta^2}\nabla_{\boldsymbol{x}_i}\|\boldsymbol{y} - \mathcal{M}(\hat{\boldsymbol{x}}_N^i(\boldsymbol{x}_i;\sigma(t_i))\|_2^2, \tag{4}$$

where $\hat{\boldsymbol{x}}_N^i := D(\boldsymbol{x}_i;\sigma(t_i))$ denotes the estimation of the final denoised data at each denoising step $i$. Applying the Baye's rule, the gradient direction of the guided diffusion is therefore:

$$\nabla_{\boldsymbol{x}_i}\log p(\boldsymbol{x}_i|\boldsymbol{y}) \approx s(\boldsymbol{x}_i) - \zeta\nabla_{\boldsymbol{x}_i}\|\boldsymbol{y} - \mathcal{M}(\hat{\boldsymbol{x}}_N^i)\|_2^2, \tag{5}$$

where $s(\boldsymbol{x}) = \nabla_{\boldsymbol{x}}\log p(\boldsymbol{x})$ is the original score function, and $\zeta = 1/\delta^2$.

### 3.3 Solving PDEs with Guided Diffusion

Our work focuses on two classes of PDEs: static PDEs and dynamic time-dependent PDEs. Static systems (e.g., Darcy Flow or Poisson equations) are defined by a time-independent function $f$:

$$f(\boldsymbol{c}; \boldsymbol{a}, \mathbf{u}) = 0 \text{ in } \Omega \subset \mathbb{R}^d, \qquad \mathbf{u}(\boldsymbol{c}) = \boldsymbol{g}(\boldsymbol{c}) \text{ in } \partial\Omega, \tag{6}$$

where $\Omega$ is a bounded domain, $\boldsymbol{c} \in \Omega$ is a spatial coordinate, $\boldsymbol{a} \in \mathcal{A}$ is the PDE coefficient field, and $\mathbf{u} \in \mathcal{U}$ is the solution field. $\partial\Omega$ is the boundary of the domain $\Omega$ and $\mathbf{u}|_{\partial\Omega} = \boldsymbol{g}$ is the boundary constraint. We aim to recover both $\boldsymbol{a}$ and $\mathbf{u}$ from sparse observations on either $\boldsymbol{a}$ or $\mathbf{u}$ or both.

Similarly, we consider the dynamic systems (e.g., Navier-Stokes):

$$\begin{aligned} f(\boldsymbol{c}, \tau; \boldsymbol{a}, \mathbf{u}) &= 0, & \text{in } \Omega \times (0, \infty) \\ \mathbf{u}(\boldsymbol{c}, \tau) &= \boldsymbol{g}(\boldsymbol{c}, \tau), & \text{in } \partial\Omega \times (0, \infty) \\ \mathbf{u}(\boldsymbol{c}, \tau) &= \boldsymbol{a}(\boldsymbol{c}, \tau), & \text{in } \bar{\Omega} \times \{0\} \end{aligned} \tag{7}$$

where $\tau$ is a temporal coordinate, $\boldsymbol{a} = \mathbf{u}_0 \in \mathcal{A}$ is the initial condition, $\mathbf{u}$ is the solution field, and $\mathbf{u}|_{\Omega \times (0,\infty)} = \boldsymbol{g}$ is the boundary constraint. We aim to simultaneously recover both $\boldsymbol{a}$ and the solution $\mathbf{u}_T := \mathbf{u}(\cdot, T)$ at a specific time $T$ from sparse observations on either $\boldsymbol{a}$, $\mathbf{u}_T$, or both.

Finally, we explore the recovery of the states across all timesteps $\mathbf{u}_{0:T}$ in 1D dynamic systems governed by Burger's equation. Our network $D_\theta$ models the distribution of all 1D states, including the initial condition $\mathbf{u}_0$ and solutions $\mathbf{u}_{1:T}$ stacked in the temporal dimension, forming a 2D dataset.

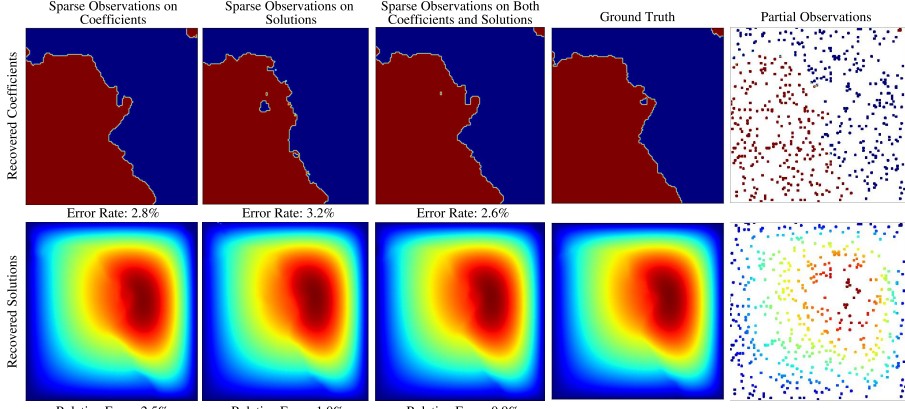

Figure 2: Different from forward and inverse PDE solvers, DiffusionPDE can take sparse observations on either the coefficient $\boldsymbol{a}$ or the solution $\mathbf{u}$ to recover both of them, using one trained network. Here, we show the recovered $\boldsymbol{a}$ and $\mathbf{u}$ of the Darcy's eqaution given sparse observations on $\boldsymbol{a}$, $\mathbf{u}$, or both. Compared with the ground truth, we see that our method successfully recovers the PDE in all cases.

**Guided Diffusion Algorithm** In the data-driven PDE literature, the above tasks can be achieved by learning directional mappings between $\boldsymbol{a}$ and $\mathbf{u}$ (or $u_T$ for dynamic systems). Thus, existing methods typically train separate neural networks for the forward solution operator $\mathcal{F} : \mathcal{A} \to \mathcal{U}$ and the inverse solution operator $\mathcal{I} : \mathcal{U} \to \mathcal{A}$.

Our method unifies the forward and inverse operators with a single network and an algorithm using the guided diffusion framework. DiffusionPDE can handle arbitrary sparsity patterns with one pre-trained diffusion model $D_\theta$ that learns the joint distribution of $\mathcal{A}$ and $\mathcal{U}$, concatenated on the channel dimension, denoted $\mathcal{X}$. Thus, our data $\boldsymbol{x} \in \mathcal{X}$, where $\mathcal{X} := \mathcal{A} \times \mathcal{U}$. We follow the typical diffusion model procedures [45] to train our model on a family of PDEs.

Once we train the diffusion model $D_\theta$, we employ our physics-informed DPS [46] formulation during inference to guide the sampling of $\boldsymbol{x} \in \mathcal{X}$ that satisfies the sparse observations and the given PDE, as detailed in Algorithm 1. We follow Eq. 5 to modify the score function using the two guidance terms:

$$\nabla_{\boldsymbol{x}_i} \log p(\boldsymbol{x}_i | \boldsymbol{y}_{obs}, f) \approx \nabla_{\boldsymbol{x}_i} \log p(\boldsymbol{x}_i) - \zeta_{obs} \nabla_{\boldsymbol{x}_i} \mathcal{L}_{obs} - \zeta_{pde} \nabla_{\boldsymbol{x}_i} \mathcal{L}_{pde}, \tag{8}$$

where $\boldsymbol{x}_i$ is the noisy data at denoising step $i$, $\boldsymbol{y}_{obs}$ are the observed values, and $f(\cdot) = \mathbf{0}$ is the underlying PDE condition. $\mathcal{L}_{obs}$ and $\mathcal{L}_{pde}$ respectively represent the MSE loss of the sparse observations and the PDE equation residuals:

$$\mathcal{L}_{obs}(\boldsymbol{x}_i, \boldsymbol{y}_{obs}; D_\theta) = \frac{1}{n} \|\boldsymbol{y}_{obs} - \hat{\boldsymbol{x}}_N^i\|_2^2 = \frac{1}{n} \sum_{j=1}^n (\boldsymbol{y}_{obs}(\boldsymbol{o}_j) - \hat{\boldsymbol{x}}_N^i(\boldsymbol{o}_j))^2,$$

$$\mathcal{L}_{pde}(\boldsymbol{x}_i; D_\theta, f) = \frac{1}{m} \|\mathbf{0} - f(\hat{\boldsymbol{x}}_N^i)\|_2^2 = \frac{1}{m} \sum_j \sum_k f(\boldsymbol{c}_j, \tau_k; \hat{\mathbf{u}}_j, \hat{\boldsymbol{a}}_j)^2, \tag{9}$$

where $\hat{\boldsymbol{x}}_N^i = D_\theta(\boldsymbol{x}_i)$ is the clean image estimate at denoising timestep $i$, which can be split into coefficient $\hat{\mathbf{u}}_i$ and solution $\hat{\boldsymbol{a}}_i$. Here, $m$ is the total number of grid points (i.e., pixels), $n$ is the number of sparse observation points. $\boldsymbol{o}_j$ represents the spatio-temporal coordinate of $j$th observation. Note that, without loss of generality, $\mathcal{L}_{pde}$ can be accumulated for all applicable PDE function $f$ in the system, and the time component $\tau_k$ is ignored for static systems.

## 4 Experiments

### 4.1 PDE Problem Settings

We show the usefulness of DiffusionPDE across various PDEs for inverse and forward problems and compare it against recent learning-based techniques. We test on the following families of PDEs.

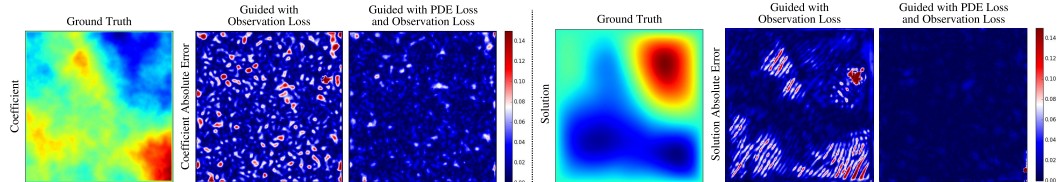

Figure 3: Usefulness of PDE loss. We visualize the absolute errors of the recovered coefficient and solution of the Helmholtz equation with and w/o PDE loss. We compare having only the observation loss with applying the additional PDE loss. The errors drop significantly when using PDE loss.

**Darcy Flow.** Darcy flow describes the movement of fluid through a porous medium. In our experiment, we consider the static Darcy Flow with a no-slip boundary $\partial\Omega$

$$-\nabla \cdot (\boldsymbol{a}(\boldsymbol{c})\nabla\mathbf{u}(\boldsymbol{c})) = q(\boldsymbol{c}), \qquad \boldsymbol{c} \in \Omega$$
$$\mathbf{u}(\boldsymbol{c}) = 0, \qquad \boldsymbol{c} \in \partial\Omega \tag{10}$$

Here the coefficient $\boldsymbol{a}$ has binary values. We set $q(\boldsymbol{c}) = 1$ for constant force. The PDE guidance function is thus $f = \nabla \cdot (\boldsymbol{a}(\boldsymbol{c})\nabla\mathbf{u}(\boldsymbol{c})) + q(\boldsymbol{c})$.

**Inhomogeneous Helmholtz Equation.** We consider the static inhomogeneous Helmholtz Equation with a no-slip boundary on $\partial\Omega$, which describes wave propagation:

$$\nabla^2\mathbf{u}(\boldsymbol{c}) + k^2\mathbf{u}(\boldsymbol{c}) = \boldsymbol{a}(\boldsymbol{c}), \qquad \boldsymbol{c} \in \Omega$$
$$\mathbf{u}(\boldsymbol{c}) = 0, \qquad \boldsymbol{c} \in \partial\Omega \tag{11}$$

The coefficient $\boldsymbol{a}$ is a piecewise constant function and $k$ is a constant. Note 11 is the Poisson equation when $k = 0$. Setting $k = 1$ for Helmholtz equations, the PDE guidance function is $f = \nabla^2\mathbf{u}(\boldsymbol{c}) + k^2\mathbf{u}(\boldsymbol{c}) - \boldsymbol{a}(\boldsymbol{c})$.

**Non-bounded Navier-Stokes Equation.** We study the non-bounded incompressive Navier-Stokes equation regarding the vorticity.

$$\partial_t w(\boldsymbol{c}, \tau) + v(\boldsymbol{c}, \tau) \cdot \nabla w(\boldsymbol{c}, \tau) = \nu\Delta w(\boldsymbol{c}, \tau) + q(\boldsymbol{c}), \quad \boldsymbol{c} \in \Omega, \tau \in (0, T]$$
$$\nabla \cdot v(\boldsymbol{c}, \tau) = 0, \qquad \boldsymbol{c} \in \Omega, \tau \in [0, T] \tag{12}$$

Here $w = \nabla \times v$ is the vorticity, $v(\boldsymbol{c}, \tau)$ is the velocity at $\boldsymbol{c}$ at time $\tau$, and $q(\boldsymbol{c})$ is a force field. We set the viscosity coefficient $\nu = 10^{-3}$ and correspondingly the Reynolds number $Re = \frac{1}{\nu} = 1000$.

DiffusionPDE learns the joint distribution of $w_0$ and $w_T$ and we take $T = 10$ which simulates 1 second. Since $T \gg 0$, we cannot accurately compute the PDE loss from our model outputs. Therefore, given that $\nabla \cdot w(\boldsymbol{c}, \tau) = \nabla \cdot (\nabla \times v) = 0$, we use simplified $f = \nabla \cdot w(\boldsymbol{c}, \tau)$.

**Bounded Navier-Stokes Equation.** We study the bounded 2D imcompressive Navier Stokes regarding the velocity $v$ and pressure $p$.

$$\partial_t v(\boldsymbol{c}, \tau) + v(\boldsymbol{c}, \tau) \cdot \nabla v(\boldsymbol{c}, \tau) + \frac{1}{\rho}\nabla p = \nu\nabla^2 v(\boldsymbol{c}, \tau), \qquad \boldsymbol{c} \in \Omega, \tau \in (0, T]$$
$$\nabla \cdot v(\boldsymbol{c}, \tau) = 0, \qquad \boldsymbol{c} \in \Omega, \tau \in (0, T]. \tag{13}$$

We set the viscosity coefficient $\nu = 0.001$ and the fluid density $\rho = 1.0$. We generate 2D cylinders of random radius at random positions inside the grid. Random turbulence flows in from the top of the grid, with the velocity field satisfying no-slip boundary conditions at the left and right edges, as well as around the cylinder $\partial\Omega_{left,right,cylinder}$. DiffusionPDE learns the joint distribution of $v_0$ and $v_T$ at $T = 4$, which simulates 0.4 seconds. Therefore, we similarly use $f = \nabla \cdot v(\boldsymbol{c}, \tau)$ as before.

**Burgers' Equation.** We study the Burgers' equation with periodic boundary conditions on a 1D spatial domain of unit length $\Omega = (0, 1)$. We set the viscosity to $\nu = 0.01$. In our experiment, the initial condition $u_0$ has a shape of $128 \times 1$, and we take 127 more time steps after the initial state to form a 2D $u_{0:T}$ of size $128 \times 128$.

$$\partial_t u(\boldsymbol{c}, \tau) + \partial_{\boldsymbol{c}}(u^2(\boldsymbol{c}, \tau)/2) = \nu\partial_{\boldsymbol{cc}}u(\boldsymbol{c}, \tau), \qquad \boldsymbol{c} \in \Omega, \tau \in (0, T]$$
$$u(\boldsymbol{c}, 0) = u_0(\boldsymbol{c}), \qquad \boldsymbol{c} \in \Omega \tag{14}$$

We can reliably compute $f = \partial_t u(\boldsymbol{c}, \tau) + \partial_{\boldsymbol{c}}(u^2(\boldsymbol{c}, \tau)/2) - \nu\partial_{\boldsymbol{cc}}u(\boldsymbol{c}, \tau)$ with finite difference since we model densely on the time dimension.

Table 1: Relative errors of solutions (or final states) and coefficients (or initial states) when solving forward and inverse problems respectively with sparse observations. Error rates are used for the inverse problem of Darcy Flow.

|  |  | DiffusionPDE | PINO | DeepONet | PINNs | FNO |
|---|---|---|---|---|---|---|
| Darcy Flow | Forward | **2.5%** | 35.2% | 38.3% | 48.8% | 28.2% |
|  | Inverse | **3.2%** | 49.2% | 41.1% | 59.7% | 49.3% |
| Poisson | Forward | **4.5%** | 107.1% | 155.5% | 128.1% | 100.9% |
|  | Inverse | **20.0%** | 231.9% | 105.8% | 130.0% | 232.7% |
| Helmholtz | Forward | **8.8%** | 106.5% | 123.1% | 142.3% | 98.2% |
|  | Inverse | **22.6%** | 216.9% | 132.8% | 160.0% | 218.2% |
| Non-bounded Navier-Stokes | Forward | **6.9%** | 101.4% | 103.2% | 142.7% | 101.4% |
|  | Inverse | **10.4%** | 96.0% | 97.2% | 146.8% | 96.0% |
| Bounded Navier-Stokes | Forward | **3.9%** | 81.1% | 97.7% | 100.1% | 82.8% |
|  | Inverse | **2.7%** | 69.5% | 91.9% | 105.5% | 69.6% |

## 4.2 Dataset Preparation and Training

We first test DiffusionPDE on jointly learning the forward mapping $\mathcal{F} : \mathcal{A} \rightarrow \mathcal{U}$ and the inverse mapping $\mathcal{I} : \mathcal{U} \rightarrow \mathcal{A}$ given sparse observations. In our experiments, we define our PDE over the unit square $\Omega = (0,1)^2$, which we represent as a $128 \times 128$ grid. We utilize Finite Element Methods (FEM) to generate our training data. Specifically, we run FNO's [3] released scripts to generate Darcy Flows and the vorticities of the Navier-Stokes equation. Similarly, we generate the dataset of Poisson and Helmholtz using second-order finite difference schemes. To add more complex boundary conditions, we use Difftaichi [57] to generate the velocities of the bounded Navier-Stokes equation. We train the joint diffusion model for each PDE on three A40 GPUs for approximately 4 hours, using 50,000 data pairs. For Burgers' equation, we train the diffusion model on a dataset of 50,000 samples produced as outlined in FNO [3]. We randomly select 5 out of 128 spatial points on $\Omega$ to simulate sensors that provide measurements across time.

## 4.3 Baseline Methods

We compare DiffusionPDE with state-of-the-art learning-based methods, including PINO [4], Deep-ONet [5], PINNs [6], and FNO [3]. However, note that none of these methods show operation on partial observations. These methods can learn mappings between $a$ and $\mathbf{u}$ or $\mathbf{u}_0$ and $\mathbf{u}_{1:T}$ with full observations, allowing them to also solve the mapping between $\mathbf{u}_0$ and $\mathbf{u}_T$. PINNs map input $a$ to output $\mathbf{u}$ by optimizing a combined loss function that incorporates both the solution $\mathbf{u}$ and the PDE residuals. DeepONet employs a branch network to encode input function values sampled at discrete points and a trunk network to handle the coordinates of the evaluated outputs. FNO maps from the parametric space to the solution space using Fourier transforms. PINO enhances FNO by integrating PDE loss during training and refining the model with PDE loss finetuning. We train all four baseline methods on both forward and inverse mappings using full observation of $a$ or $\mathbf{u}$ for both static and dynamic PDEs. We tried training the baseline models on partial observations, but we noticed degenerate training outcomes (see supplementary for details). Overall, they are intended for *full observations* and may not be suitable for sparse measurements.

More closely related to our method, GraphPDE [8] demonstrates the ability to recover the initial state using sparse observations on the final state, a task that other baselines struggle with. Therefore, we compare against GraphPDE for the inverse problem of bounded Navier-Stokes (NS) equation, which is the setup used in their report. GraphPDE uses a trained latent space model and a bounded forward GNN model to solve the inverse problem with sparse sensors and thus is incompatible with unbounded Navier-Stokes. We create bounded meshes using our bounded grids to train the GNN model and train the latent prior with $v_{0:T}$ for GraphPDE.

While we employ guided sampling to reconstruct the solutions, Classifier-Free Guidance (CFG) [58] offers an alternative approach where the diffusion model is conditioned on sparse input data. Shu et al. [54] extend this method by developing an optimized CFG approach that conditions on the PDE loss,

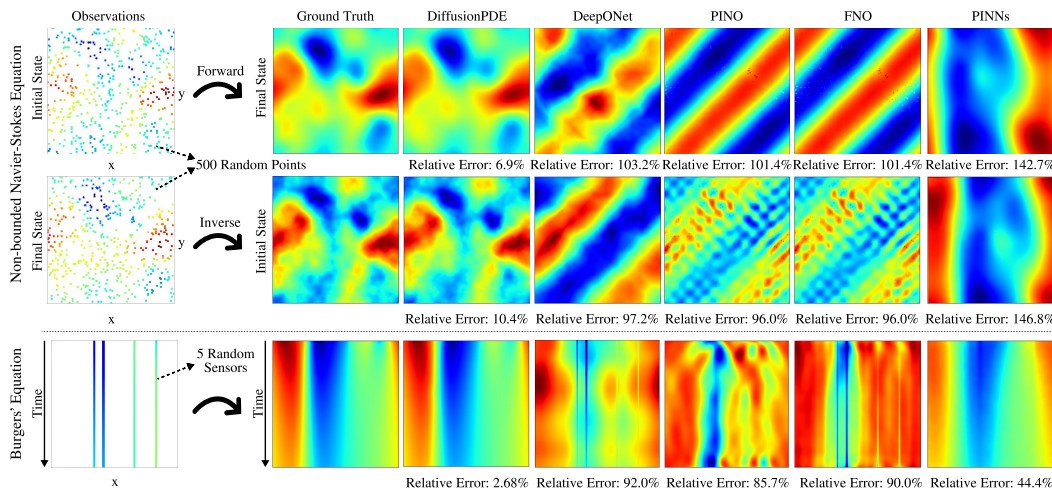

Figure 4: We compare DiffusionPDE with state-of-the-art neural PDE solvers [3–6]. In the forward Navier-Stokes problem, we give 500 sparse observations of the initial state to solve for the final state. In the inverse set-up, we take observations of the final state and solve for the initial. For the Burgers' equation, we use 5 sensors throughout all time steps and want to recover the solution at all time steps. Note that we train on neighboring snapshot pairs for the baselines in order to add continuous observations of the Burgers' equation. Results show that existing methods do not support PDE solving under sparse observations, and we believe they are not easily extendable to do so. We refer readers to the supplementary for a complete set of visual results.

using the observation as a low-resolution input. Additionally, OFormer [59] is another model designed to reconstruct the full solution using transformers, offering a shorter inference runtime. Consequently, we compare our approach against these methods for solving the unbounded Navier-Stokes equation.

## 4.4 Main Evaluation Results

We respectively address the forward problem and the inverse problem with sparse observations of $a$ or $\mathbf{u}$. For the forward problem, we randomly select coefficients (initial states) as sparse observations and then compare the predicted solutions (final states) with the ground truth. Specifically, we select 500 out of $128 \times 128$ points, approximately 3%, on the coefficients of Darcy Flow, Poisson equation, Helmholtz equation, and the initial state of the non-bounded Navier-Stokes equation. For the bounded Navier-Stokes equation, we use 1% observed points beside the boundary of the cylinder in 2D. Similarly, for the inverse problem, we randomly sample points on solutions (final states) as sparse observations, using the same number of observed points as in the forward model for each PDE.

We show the relative errors of all methods regarding both forward and inverse problems in Table 1. Since the coefficients of Darcy Flow are binary, we evaluate the error rates of our prediction. Non-binary data is evaluated using mean pixel-wise relative error. We report error numbers averaged across 1,000 random scenes and observations for each PDE. DiffusionPDE outperforms other methods including PINO [4], DeepONet [5], PINNs [6], and FNO [3] for both directions with sparse observations, demonstrating the novelty and uniqueness of our approach. For the inverse problems of the Poisson and Helmholtz equations, DiffusionPDE exhibits higher error rates due to the insufficient constraints within the coefficient space, produced from random fields. In Fig. 4, we visualize the results for solving both the forward and inverse problem of the non-bounded Navier-Stokes. We refer to the *supplementary* for additional visual results. While other methods may produce partially correct results, DiffusionPDE outperforms them and can recover results very close to the ground truth.

For the inverse problem of the bounded Navier-Stokes equation, we further compare DiffusionPDE with GraphPDE, as illustrated in Fig. 5. Our findings reveal that DiffusionPDE surpasses GraphPDE [8] in accuracy, reducing the relative error from 12.0% to 2.7% with only 1% observed points.

We further show whether DiffusionPDE can jointly recover both $a$ and $\mathbf{u}$ by analyzing the retrieved $a$ and $\mathbf{u}$ with sparse observations on different sides as well as on both sides. In Fig. 2, we recover the coefficients and solutions of Darcy Flow by randomly observing 500 points on only coefficient space, only space solution space, and both. Both coefficients and solutions can be recovered with low errors

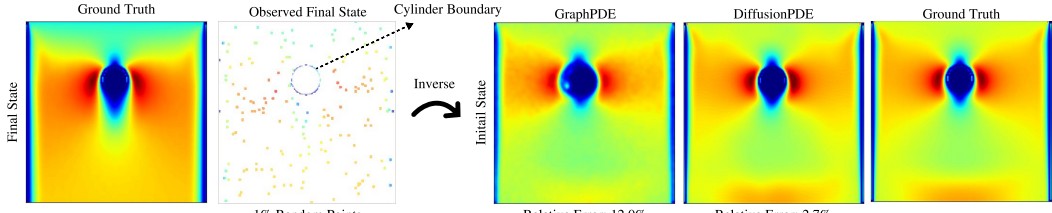

Figure 5: We compare GraphPDE [8] and our method for solving the inverse bounded Navier-Stokes equation. Given the boundary conditions and $1\%$ observations of the final vorticity field, we solve the initial vorticity field. We set the fliuds to flow in from the top, with boundary conditions at the edges and a middle cylinder. While GraphPDE can recover the overall pattern of the initial state, it suffers from noise when the fluid passes the cylinder and misses the high vorticities at the bottom.

for each situation. We therefore conclude that DiffusionPDE can solve the forward problem and the inverse problem simultaneously with sparse observations at any side without retraining our network.

## 4.5 Advantage of Guided Sampling

To demonstrate the clear advantage of our guided sampling method, we evaluate both the forward and inverse processes of the unbounded Navier-Stokes equation, comparing our DiffusionPDE approach with Diffusion using CFG when considering only the initial and final states given 500 observation points, as illustrated in Fig. 6. Our DiffusionPDE method consistently achieves lower relative errors across both evaluations.

Furthermore, in Fig. 7, we compare our results with those of Shu et al. [54], where the full time intervals are solved autoregressively using an optimized CFG method. In their approach, the error in the final state increases to approximately 13%, which is notably higher than that of our two-state model. Additionally, the relative errors of the transformer-based approach, OFormer [59], are around 17% and 23%, which are significantly larger than those observed with DiffusionPDE.

## 4.6 Recovering Solutions Throughout a Time Interval

We demonstrate that DiffusionPDE is capable of retrieving all time steps throughout the time interval $[0, T]$ from continuous observations on sparse sensors. To evaluate its ability to recover $u_{0:T}$ with sparse sensors, we study the 1D dynamic Burgers' equation, where DiffusionPDE learns the distribution of $u_{0:T}$ using a 2D diffusion model. To apply continuous observation on PINO, DeepONet, FNO, and PINNs, we train them on neighboring snapshot pairs. Our experiment results in a test relative error of 2.68%, depicted in Fig. 4, which is significantly lower than other methods.

## 4.7 Additional Analysis

We examine the effects of different components of our algorithm such as PDE loss and observation samplings. We strongly encourage readers to view the supplementary for more details of these analyses as well as additional experiments.

**PDE Loss.** To verify the role of the PDE guidance loss of Eq. 8 during the denoising process, we visualize the errors of recovered $a$ and $u$ of Helmholtz equation with or without PDE loss. Here, we run our DPS algorithm with 500 sparse observed points on both the coefficient $a$ and solution $u$ and study the effect of the additional PDE loss guidance. The relative error of $u$ reduces from 9.3% to 0.6%, and the relative error of $a$ reduces from 13.2% to 9.4%. Therefore, we conclude that PDE guidance helps smooth the prediction and improve the accuracy.

**Number of Observations.** We examine the results of DiffusionPDE in solving forward and inverse problems when there are 100, 300, 500, and 1000 random observations on $a$, $u$, or both $a$ and $u$. The error of DiffusionPDE decreases as the number of sparse observations increases. DiffusionPDE is capable of recovering both $a$ and $u$ with errors $1\% \sim 10\%$ with approximately $6\%$ observation points at any side for most PDE families. DiffusionPDE becomes insensitive to the number of observations and can solve the problems well once more than $3\%$ of the points are observed.

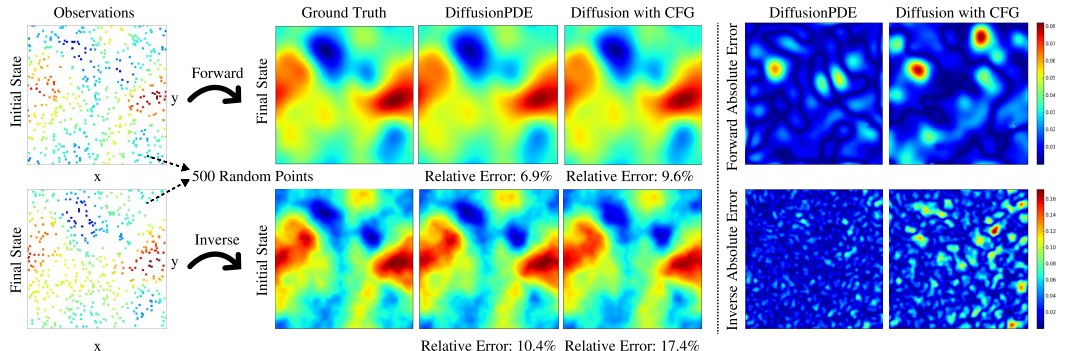

Figure 6: We compare the performance of DiffusionPDE and Diffusion with CFG for the unbounded Navier-Stokes equation, and visualize the error. With 500 observation points, DiffusionPDE demonstrates superior accuracy, achieving lower errors in both forward and inverse problem-solving.

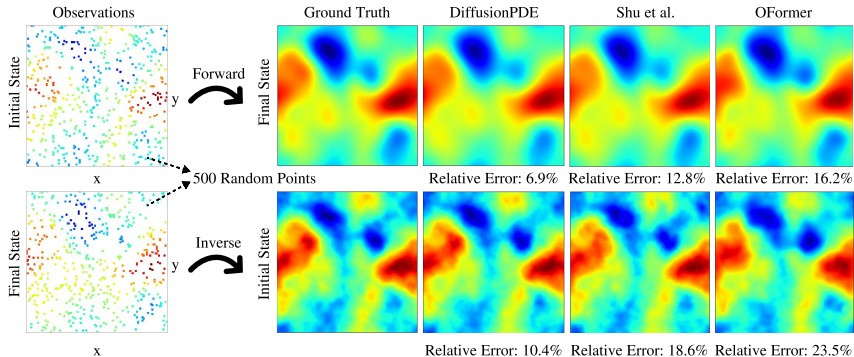

Figure 7: We compare our DiffusionPDE method with the approaches of Shu et al. [54] and OFormer [59] for the unbounded Navier-Stokes equation. Using 500 observation points, DiffusionPDE effectively solves both the forward and inverse problems, achieving significantly lower errors.

**Observation Sampling Pattern.** While CFG struggles with robustness, we show that DiffusionPDE is robust to different sampling patterns of the sparse observations, including grid and non-uniformly concentrated patterns. Note that even when conditioned on the full observations, our approach performs on par with the current best methods, likely due to the inherent resilience of our guided diffusion algorithm. Additionally, DiffusionPDE can leverage continuous coordinates with bilinear interpolation in the prediction space to obtain predicted values for points that do not lie directly on the grid, without compromising accuracy.

## 5    Conclusion and Future Work

In this work, we develop DiffusionPDE, a diffusion-based PDE solver that addresses the challenge of solving PDEs from partial observations by filling in missing information using generative priors. We formulate a diffusion model that learns the joint distribution of the coefficient (or initial state) space and the solution (or final state) space. During the sampling process, DiffusionPDE can flexibly generate plausible data by guiding its denoising with sparse measurements and PDE constraints. Our new approach leads to significant improvements over existing state-of-the-art methods, advancing toward a general PDE-solving framework that leverages the power of generative models.

Several promising directions for future research have emerged from this work. Currently, DiffusionPDE is limited to solving slices of 2D dynamic PDEs; extending its capabilities to cover full time intervals of these equations presents a significant opportunity. Moreover, the model's struggle with accuracy in spaces that lack constraints is another critical area for exploration. DiffusionPDE also suffers from a slow sampling procedure, and a faster solution might be desired.

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

# Appendix

## A  Overview

In this supplementary material, we provide additional details to complement the main paper. Section B elaborates on the data generation process. Section C outlines the sampling implementation, and Section D highlights error reductions achieved by integrating PDE loss. Section E presents comprehensive visual results for both forward and inverse computations using sparse observations, which are not included in the main text. In Section F, we discuss results from full observation scenarios across all methods. Section G justifies our decision to train the baselines on complete observation data, while Section H shows results from optimized baseline methods. Section I and J provide standard deviation and runtime analyses, and Section K examines the model's robustness against random noise and varying observation locations, as well as the stochasticity of the model. Section L and M explores how different observation numbers and resolutions affect result accuracy, offering further insight into the model's performance under varying conditions. Lastly, Section N compares DiffusionPDE with additional baseline methods, including RBF kernel and U-Net.

## B  Data Generation Details

We generate 50,000 samples for each PDE and all diffusion models are trained on Nvidia A40 GPUs.

### B.1  Static PDEs

We derived the methods of data generation for static PDEs from [3]. We first generate Gaussian random fields on $(0,1)^2$ so that $\mu \sim \mathcal{N}(0, (-\Delta + 9\mathbf{I})^{-2})$. For Darcy Flow, we let $a = f(\mu)$ so that:

$$\begin{cases} a(x) = 12, & \text{if } \mu(x) \geq 0 \\ a(x) = 3, & \text{if } \mu(x) < 0 \end{cases}$$

For the Poisson equation and Helmholtz equation, we let $a = \mu$ as the coefficients. We then use second-order finite difference schemes to solve the solution $u$ and enforce the no-slip boundary condition for solutions by multiplying a mollifier $\sin(\pi x_1)\sin(\pi x_2)$ for point $x = (x_1, x_2) \in (0,1)^2$. Both $a$ and $u$ have resolutions of $128 \times 128$.

### B.2  Non-bounded Navier-Stokes Equation

We derived the method to generate non-bounded Navier-Stokes equation from [3]. The initial condition $w_0$ is generated by Gaussian random field $\mathcal{N}(0, 7^{1.5}(-\Delta + 49\mathbf{I})^{-2.5})$. The forcing function follows the fixed pattern for point $(x_1, x_2)$:

$$q(x) = \frac{1}{10}(\sin(2\pi(x_1 + x_2)) + \cos(2\pi(x_1 + x_2)))$$

We then use the pseudo-spectral method to solve the Navier-Stokes equations in the stream-function formulation. We transform the equations into the spectral domain using Fourier transforms, solving the vorticity equation in the spectral domain, and then using inverse Fourier transforms to compute nonlinear terms in physical space. We simulate for 1 second with 10 timesteps, and $w_t$ has a resolution of $128 \times 128$.

### B.3  Bounded Navier-Stokes Equation

We use Difftaichi [57] to generate data for the bounded Navier-Stokes equation. Specifically, we apply the Marker-and-Cell (MAC) method by solving a pressure-Poisson equation to enforce incompressibility and iterating through predictor and corrector steps to update the velocity and pressure fields. The grid is of the resolution $128 \times 128$ and the center of the cylinder is at a random location in $[30, 60] \times [30, 90]$ with a random radius in $[5, 20]$. The fluid flows into the grid from the upper boundary with a random initial vertical velocity in $[0.5, 3]$. We simulate for 1 second with 10 timesteps and study steps 4 to 8 when the turbulence is passing the cylinder.

### B.4  Burgers' Equation

We derived the method to generate Burgers' equation from [3]. The initial condition $u_0$ is generated by Gaussian random field $\mathcal{N}(0, 625(-\Delta + 25\mathbf{I})^{-2})$. We solve the PDE with a spectral method and simulate 1 second with 127 additional timesteps. The final $u_{0:T}$ space has a resolution of $128 \times 128$.

## C  Guided Sampling Details

For experiments with sparse observations or sensors, we find that DiffusionPDE performs the best when weights $\zeta$ are selected as shown in Table 2. During the initial 80% of iterations in the sampling process, guidance is exclusively provided by the observation loss $\mathcal{L}_{obs}$. Subsequently, after 80% of the iterations have been completed, we introduce the PDE loss $\mathcal{L}_{pde}$, and reduce the weighting factor $\zeta_{obs}$ for the observation loss, by a factor of 10. This adjustment shifts the primary guiding influence to the PDE loss, thereby aligning the diffusion model more closely with the dynamics governed by the partial differential equations.

Table 2: The weights assigned to the PDE loss and the observation loss vary depending on whether the observations pertain to the coefficients (or initial states) $a$ or to the solutions (or final states) $u$.

| | | Darcy Flow | Poisson | Helmholtz | Non-bounded Navier-Stokes | Bounded Navier-Stokes | Burgers' equation |
|---|---|---|---|---|---|---|---|
| $\zeta_{obs}$ | $a$ | $2.5 \times 10^3$ | $4 \times 10^2$ | $2 \times 10^2$ | $5 \times 10^2$ | $2.5 \times 10^2$ | $3.2 \times 10^2$ |
| | $u$ | $10^6$ | $2 \times 10^4$ | $3 \times 10^4$ | $5 \times 10^2$ | $2.5 \times 10^2$ | - |
| $\zeta_{pde}$ | | $10^3$ | $10^2$ | $10^2$ | $10^2$ | $10^2$ | $10^2$ |

## D  Improvement in Prediction through PDE Loss Term

DiffusionPDE performs better when we apply the PDE loss term $\mathcal{L}_{pde}$ in addition to the observation loss term $\mathcal{L}_{obs}$ as guidance, as shown in Table 3. The errors in both the coefficients ( initial states) $a$ and the solutions (final states) $u$ significantly decrease. We also visualize the recovered $a$ and $u$ and corresponding absolute errors of Darcy Flow, Poisson equation, and Helmholtz equation in Fig. 8. It is demonstrated that the prediction becomes more accurate with the combined guidance of PDE loss and observation loss than with only observation loss.

Table 3: DiffusionPDE' prediction errors of coefficients (initial states) $a$ and solutions (final states) $u$ with sparse observation on both $a$ and $u$, guided by different loss functions.

| Loss Function | Side | Darcy Flow | Poisson | Helmholtz | Non-bounded Navier-Stokes | Bounded Navier-Stokes |
|---|---|---|---|---|---|---|
| $\mathcal{L}_{obs}$ | $a$ | 4.6% | 12.1% | 13.2% | 8.2% | 6.4% |
| | $u$ | 4.8% | 6.5% | 9.3% | 7.6% | 3.3% |
| $\mathcal{L}_{obs} + \mathcal{L}_{pde}$ | $a$ | 3.4% | 10.3% | 9.4% | 4.9% | 1.7% |
| | $u$ | 1.7% | 0.3% | 0.6% | 0.6% | 1.4% |

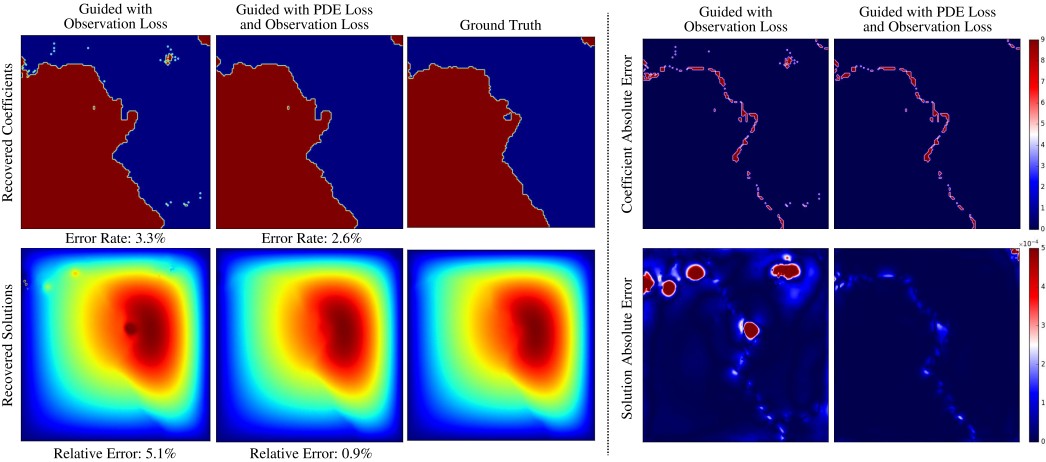

(a) Recovered coefficients, solutions, and corresponding absolute errors of Darcy Flow.

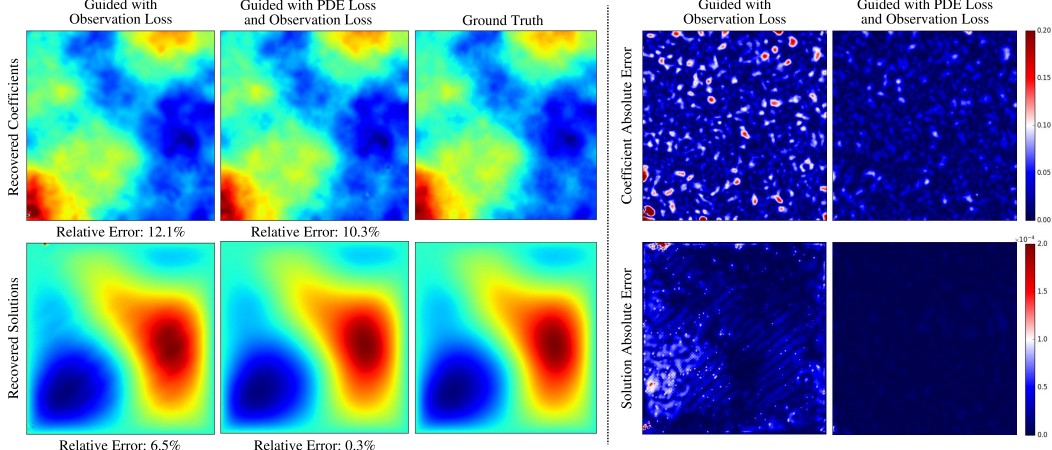

(b) Recovered coefficients, solutions, and corresponding absolute errors of Poisson equation.

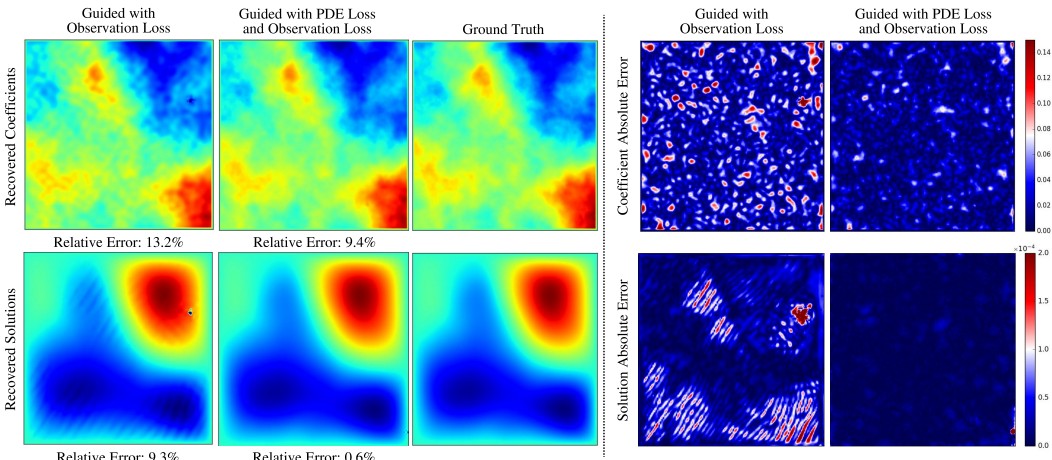

(c) Recovered coefficients, solutions, and corresponding absolute errors of Helmholtz equation.

Figure 8: Recovered coefficients, solutions, and their corresponding visualized absolute errors for various PDE families.

# E  Additional Results on All PDEs with Sparse Observation

We present the recovered results of another Burgers' equation in Fig. 9. DiffusionPDE outperforms all other methods with 5 sensors for continuous observation. We also present the recovered results for both the forward and inverse problems of all other PDEs with sparse observations, as shown in Fig. 10. Specifically, we solve the forward and inverse problems for the Darcy Flow, Poisson equation, Helmholtz equation, and non-bounded Navier-Stokes equation using 500 random points observed in either the solution space or the coefficient space. Additionally, for the bounded Navier-Stokes equation, we observe 1% of the points in the velocity field. Our findings indicate that DiffusionPDE outperforms all other methods, providing the most accurate solutions.

**Additional Data Setting for Darcy Flow**    To further demonstrate the generalization capability of our model, we conducted additional tests on different data settings for Darcy Flow. In Fig. 11, we solve the forward and inverse problems of Darcy Flow with 500 observation points, adjusting the binary values of $a$ to 20 and 16 instead of the original 12 and 3 in Section B, i.e.,

$$\begin{cases} a(x) = 20, & \text{if } \mu(x) \geq 0 \\ a(x) = 16, & \text{if } \mu(x) < 0 \end{cases}$$

Our results indicate that DiffusionPDE performs equally well under these varied data settings, showcasing its robustness and adaptability.

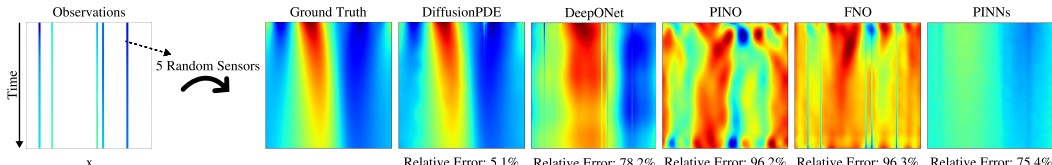

Figure 9: Results of another Burgers' equation recovered by 5 sensors throughout the time interval.

(a) Forward and inverse results of Darcy Flow recovered by 500 observation points.

(b) Forward and inverse results of Poisson equation recovered by 500 observation points.

(c) Forward and inverse results of Helmholtz equation recovered by 500 observation points.

(d) Forward and inverse results of another non-bounded Navier-Stokes equation recovered by 500 observation points.

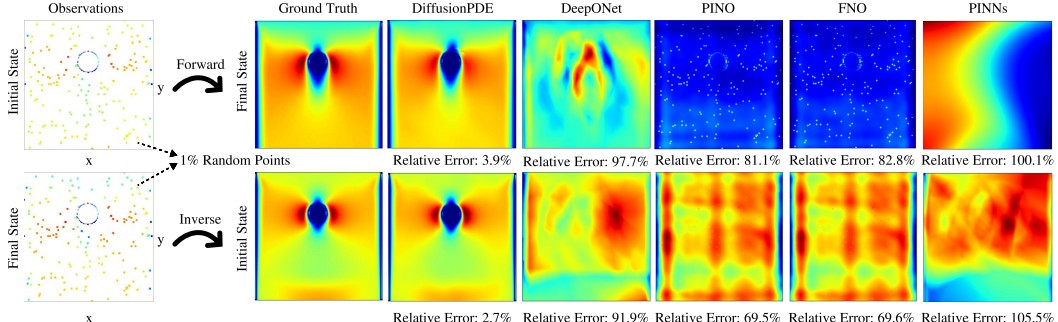

(e) Forward and inverse results of bounded Navier-Stokes equation recovered by 1% observation points.

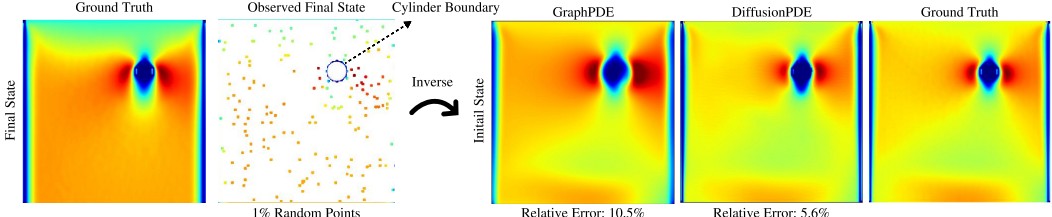

(f) Inverse results of DiffusionPDE and GraphPDE of another bounded Navier-Stokes equation recovered by 1% observation points and the known boundary of the cylinder.

Figure 10: Results of forward and inverse problems for different PDE families with sparse observation.

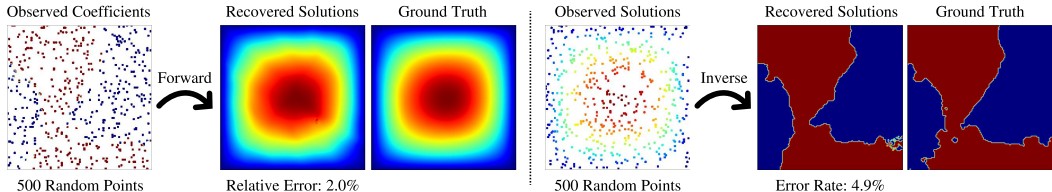

Figure 11: Forward and inverse results of Darcy Flow recovered by 500 observation points under a different data setting.

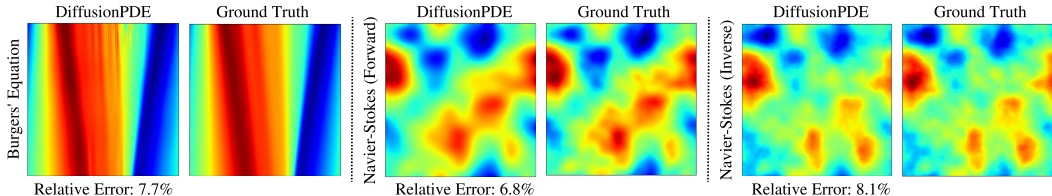

Figure 12: Results of Navier-Stokes equation and Burgers' equation with 10 times smaller viscosity.

**Additional Data Setting for Non-bounded Navier-Stokes Equation and Burgers' Equation**    We also test DiffusionPDE on the Burgers' equation with a viscosity of $1 \times 10^{-3}$ and on the non-bounded Navier-Stokes equation with a viscosity of $1 \times 10^{-4}$, which are 10 times smaller than the ones in the main paper, as shown in Fig. 12. For the Burgers' equation, we are able to recover the full time interval with 5 fixed sensors at a relative error of approximately $6\%$, which is close to the error of approximately $2 \sim 5\%$ in the main paper. For the Navier-Stokes equation, we can solve the forward and inverse problems with relative errors of approximately $7\%$ and $9\%$, respectively, using 500 observation points. The errors are also close to the ones in the main paper, where the forward and inverse errors of Navier-Stokes equation are approximately $7\%$ and $10\%$.

## F    Solving Forward and Inverse Problems with Full Observation

We have also included the errors of all methods when solving both the forward and inverse problems with full observation, as displayed in Table 4.

Table 4: Relative errors of solutions (or final states) and coefficients (or initial states) when solving forward and inverse problems with full observations. Error rates are used for the inverse problem of Darcy Flow.

|  |  | DiffusionPDE | PINO | DeepONet | PINNs | FNO |
|---|---|---|---|---|---|---|
| Darcy Flow | Forward | **2.2%** | 4.0% | 12.3% | 15.4% | 5.3% |
|  | Inverse | **2.0%** | 2.1% | 8.4% | 10.1% | 5.6% |
| Poisson | Forward | **2.7%** | 3.7% | 14.3% | 16.1% | 8.2% |
|  | Inverse | **9.8%** | 10.2% | 29.0% | 28.5% | 13.6% |
| Helmholtz | Forward | **2.3%** | 4.9% | 17.8% | 18.1% | 11.1% |
|  | Inverse | **4.0%** | 4.9% | 28.1% | 29.2% | 5.0% |
| Non-bounded Navier-Stokes | Forward | 6.1% | **1.1%** | 25.6% | 27.3% | 2.3% |
|  | Inverse | 8.6% | **6.8%** | 19.6% | 27.8% | 6.8% |
| Bounded Navier-Stokes | Forward | **1.7%** | 1.9% | 13.3% | 18.6% | 2.0% |
|  | Inverse | **1.4%** | 2.9% | 6.1% | 7.6% | 3.0% |

In general, DiffusionPDE and PINO outperform all other methods, and DiffusionPDE performs the best for all static PDEs. DiffusionPDE is capable of solving both forward and inverse problems with errors of less than 10% for all classes of discussed PDEs and is comparable to the state-of-the-art. Results of all methods regarding Darcy Flow and non-bounded Navier-Stokes equation are included in Fig. 13.

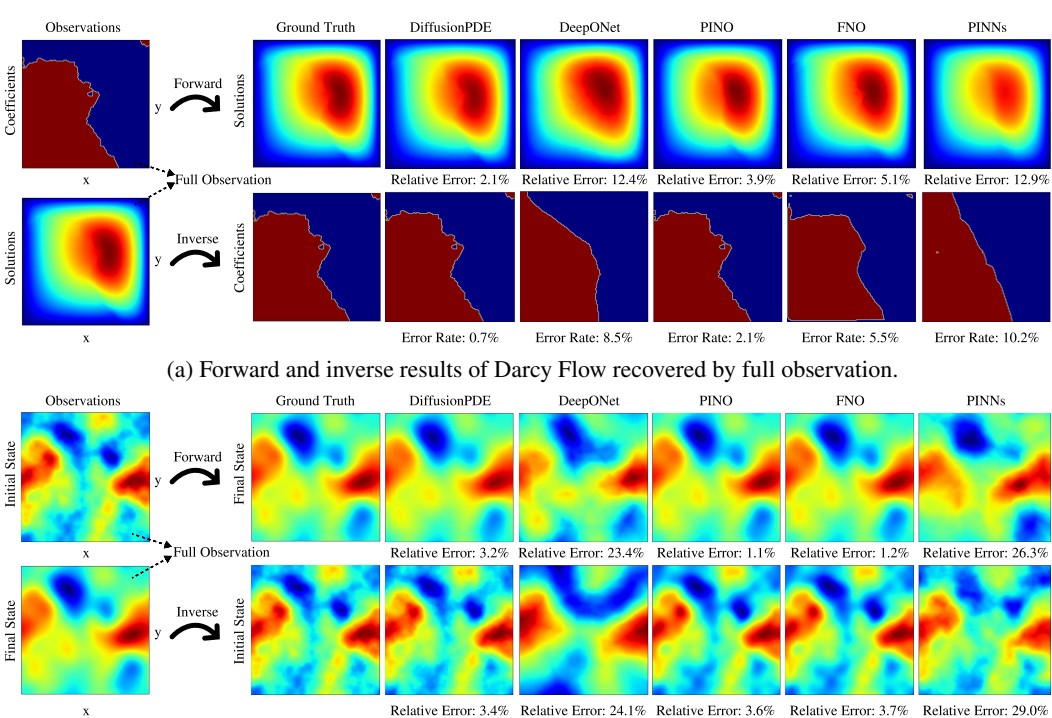

(a) Forward and inverse results of Darcy Flow recovered by full observation.

(b) Forward and inverse results of non-bounded Navier-Stokes equation recovered by full observation.

Figure 13: Results of forward and inverse problems for different PDE families with full observation.

## G    Training Baselines Methods on Partial Inputs

For our main experiments, we opt to train the baseline models (PINO, DeepONet, PINNs, FNO) on full observations for several compelling reasons: First, physics-informed models such as PINNs and PINO are unable to effectively compute the PDE loss when only sparse observations are available. Second, other models like DeepONet and FNO perform poorly with sparse observations. For instance,

Table 5: Relative errors of solutions (or final states) and coefficients (or initial states) when solving forward and inverse problems respectively with sparse observations after optimizing the baselines. Error rates are used for the inverse problem of Darcy Flow.

|  |  | DiffusionPDE | DeepONet | PINO | FNO | PINNs |
|---|---|---|---|---|---|---|
| Darcy Flow | Forward | **2.5%** | 31.3% | 32.6% | 27.8% | 6.9% |
|  | Inverse | **3.2%** | 41.1% | 49.2% | 49.3% | 59.7% |
| Poisson | Forward | **4.5%** | 73.6% | 79.1% | 70.5% | 77.8% |
|  | Inverse | **20.0%** | 75.0% | 115.0% | 118.5% | 73.9% |
| Helmholtz | Forward | **8.8%** | 77.6% | 67.7% | 84.8% | 79.2% |
|  | Inverse | **22.6%** | 100.7% | 125.3% | 131.6% | 103.7% |
| Non-bounded Navier-Stokes | Forward | **6.9%** | 96.5% | 93.3% | 91.6% | 106.1% |
|  | Inverse | **10.4%** | 71.9% | 87.8% | 89.3% | 108.6% |
| Bounded Navier-Stokes | Forward | **3.9%** | 89.1% | 80.8% | 81.2% | 84.4% |
|  | Inverse | **2.7%** | 88.6% | 47.3% | 48.7% | 82.1% |

training the DeepONet model on 500 uniformly random points for each training sample in the context of the forward problem of Darcy Flow leads to testing outcomes that are consistently similar, as illustrated in Fig. 14, regardless of the testing input. This pattern suggests that the model tends to generate a generalized solution that minimizes the average error across all potential solutions rather than converging based on specific samples. Furthermore, the partial-input-trained model exhibits poor generalization when faced with a different distribution of observations from training, indicating that it lacks flexibility—a critical attribute of our DiffusionPDE.

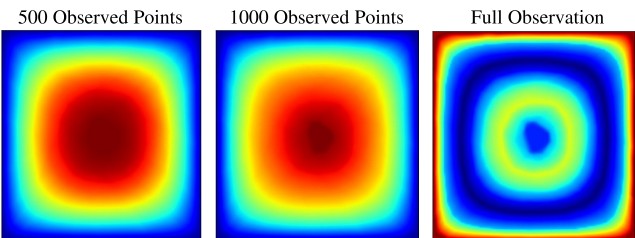

Figure 14: Predicted solutions obtained using the DeepONet model trained with 500 observation points across different numbers of observation points.

## H Baseline Optimization

We further refine the noisy outputs generated by baseline methods such as DeepONet, PINO, FNO, and PINNs. Specifically, given a partially observed parameter $a$ for the PDE $f(c; a, u) = 0$ and a pre-trained forward operator $\mathcal{F}'$, we address the problem by solving the optimization equation:

$$\min_a \mathcal{L}_{pde}(a, \mathcal{F}'(a); f) \tag{15}$$

and the results are shown in Table 5 and Fig. 15. Optimization reduces errors and smooths the solutions. However, the resulting values are smaller due to the smoothing effect from minimizing PDE loss, and the overall error compared to the ground truth remains much higher than DiffusionPDE. This may be due to the difficulty in optimizing the derivatives of noisy $a$ and $u$.

## I Standard Deviation of DiffusionPDE Experiment Results

We further assess the statistical significance of our DiffusionPDE by analyzing the standard deviations for forward and inverse problems under conditions of 500 sparse observation points and full observation, respectively, as detailed in Table 6. We evaluate our model using test sets comprising 1,000 samples for each PDE. Our findings confirm that full observation enhances the stability of

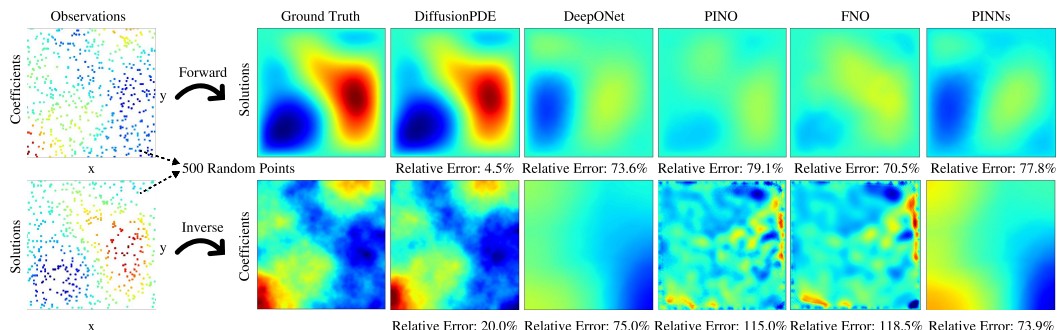

Figure 15: Results of Poisson equation after optimizing baseline methods.

the results, a predictable outcome as variability diminishes with an increase in observation points. The standard deviations are notably higher for more complex PDEs, such as the inverse problems of the Poisson and Helmholtz equations, reflecting the inherent challenges associated with these computations. Overall, DiffusionPDE demonstrates considerable stability, evidenced by relatively low standard deviations across various tests.

Table 6: Standard deviation of DiffusionPDE when solving forward and inverse problems with sparse or full observations.

|  |  | Sparse Observations | Full Observations |
|---|---|---|---|
| Darcy Flow | Forward | $2.5 \pm 0.7\%$ | $2.2 \pm 0.1\%$ |
|  | Inverse | $3.2 \pm 0.9\%$ | $2.0 \pm 0.1\%$ |
| Poisson | Forward | $4.5 \pm 0.9\%$ | $2.7 \pm 0.1\%$ |
|  | Inverse | $20.0 \pm 1.8\%$ | $9.8 \pm 0.7\%$ |
| Helmholtz | Forward | $8.8 \pm 1.0\%$ | $2.3 \pm 0.1\%$ |
|  | Inverse | $22.6 \pm 1.7\%$ | $4.0 \pm 0.6\%$ |
| Non-bounded Navier-Stokes | Forward | $6.9 \pm 0.9\%$ | $6.1 \pm 0.2\%$ |
|  | Inverse | $10.4 \pm 1.0\%$ | $8.6 \pm 0.3\%$ |
| Bounded Navier-Stokes | Forward | $3.9 \pm 0.2\%$ | $1.7 \pm 0.1\%$ |
|  | Inverse | $2.7 \pm 0.2\%$ | $1.4 \pm 0.1\%$ |

## J   Runtime Analysis

We evaluate the computing cost during the inference stage by testing a single data point on a single A40 GPU for the Navier-Stokes equation, as shown in Table 7. DiffusionPDE has a lower computing cost compared to Shu et al. [54], which autoregressively solves the full time interval. This advantage becomes more significant when we increase the number of time steps.

Table 7: Inference computing cost of sparse-observation-based methods.

| Method | DiffusionPDE (Ours) | GraphPDE | Shu et al. (2023) | OFormer |
|---|---|---|---|---|
| #Parameter (M) | 54 | 1.3 | 63 | 1.6 |
| Inference time (s) | 140 | 84 | 180 | 3.2 |
| GPU memory (GB) | 6.8 | 3.6 | 7.2 | 0.1 |

Further, we evaluate the inference runtimes on one single A40 GPU of vanilla full-observation-based methods and also the optimization time of them during the inference as introduced in Appendix H. The optimization runtimes are significantly slower, especially when using Fourier transforms.

Table 8: Average inference runtimes (in seconds) of full-observation-based methods with and without optimization.

| Method | PINO | FNO | DeepONet | PINNs |
|---|---|---|---|---|
| Vanilla | 1.0e0 | 9.8e-1 | 7.4e-1 | 1.5e0 |
| With Optimization | 6.7e2 | 6.7e2 | 3.5e1 | 3.7e1 |

# K  Robustness of DiffusionPDE

We find that DiffusionPDE is robust against sparse noisy observation. In Fig. 16, we add Gaussian noise to the 500 observed points of Darcy Flow coefficients. Our DiffusionPDE can maintain a relative error of around 10% with a 15% noise level concerning the forward problem, and the recovered solutions are shown in Fig. 17. Baseline methods such as PINO also exhibit robustness against random noise under sparse observation conditions; this is attributed to their limited applicability to sparse observation problems, leading them to address the problem in a more randomized manner.

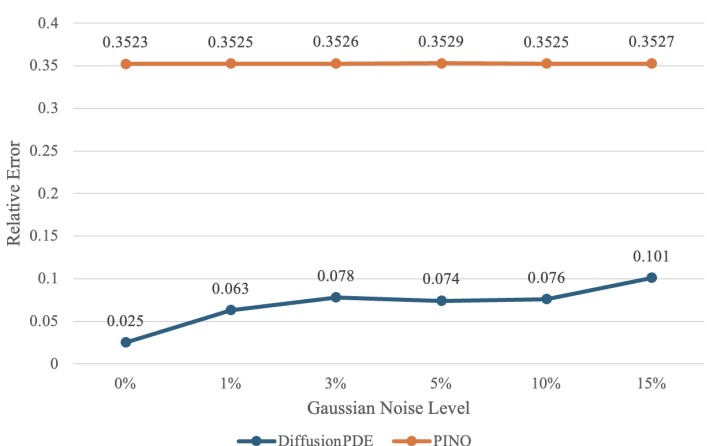

Figure 16: Relative errors of recovered Darcy Flow solutions with sparse noisy observation.

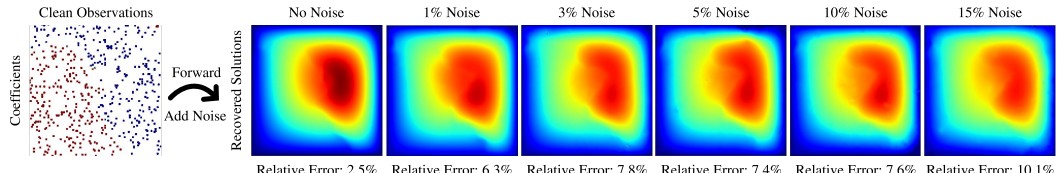

Figure 17: Recovered solutions for Darcy Flow with noisy observations.

**Robustness on Sampling Patterns**   Moreover, as mentioned in the main document, we investigate the robustness of DiffusionPDE on different sampling patterns of the observation points. Here, we address the forward problem of Darcy Flow using 500 observed coefficient points, which are non-uniformly concentrated on the left and right sides or are regularly distributed across the grid, as depicted in Fig. 18. Our results demonstrate that DiffusionPDE flexibly solves problems with arbitrary sparse observation locations within the spatial domain, without re-training the neural network model. However, the CFG method faces challenges when solving with varying sampling patterns, as demonstrated in Fig. 19. In this figure, we compare the reconstruction results of DiffusionPDE and Diffusion with CFG for the unbounded Navier-Stokes equation, where all observation points are located on the left side of the grid. The CFG approach struggles with this asymmetric sampling pattern, while DiffusionPDE maintains more accurate reconstructions.

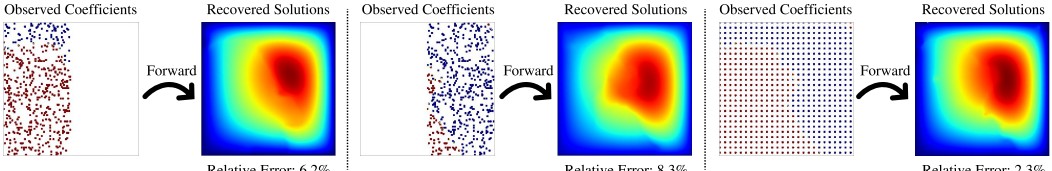

Figure 18: Recovered solutions for Darcy Flow with observations sampled using non-uniform distributions.

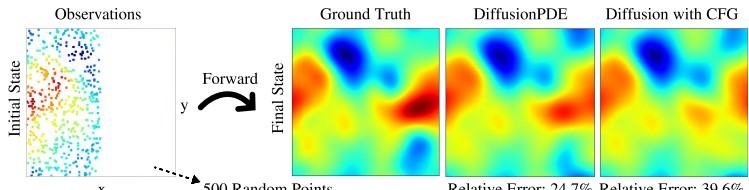

Figure 19: Comparison between DiffusionPDE and Diffusion CFG under different sampling patterns for non-bounded Navier-Stokes equation.

**Stochasticity Evaluation** Since we employ a deterministic diffusion model, with partial observations as input, the only source of stochasticity or uncertainty in our approach arises from the initial random noise. To examine this, we conducted experiments to assess the impact of different noise seeds on both the initial and final states of the Navier-Stokes equations, as demonstrated in Fig. 20. Our findings indicate that the diffusion model exhibits some degree of uncertainty in its predictions, despite the deterministic nature of the underlying framework.

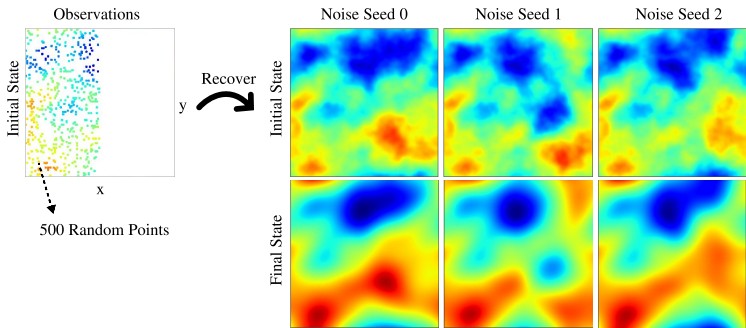

Figure 20: Different predictions of DiffusionPDE generated by different initial noise for non-bounded Navier-Stokes equation.

## L Solving Forward and Inverse Problems with Different Numbers of Observations

We also investigate how our DiffusionPDE handles varying degrees of sparse observation. Experiments are conducted on the Darcy Flow, Poisson equation, Helmholtz equation, and non-bounded Navier-Stokes equation. We examine the results of DiffusionPDE in solving forward and inverse problems when there are 100, 300, 500, and 1000 random observations on $a$, $u$, or both $a$ and $u$, as shown in Fig. 21. We have observed that the error of DiffusionPDE decreases as the number of sparse observations increases. Overall, we recover $u$ better than $a$. DiffusionPDE can recover $u$ with approximately 2% observation points at any side pretty well. DiffusionPDE is also capable of recovering both $a$ and $u$ with errors $1\% \sim 10\%$ with approximately 6% observation points at any side for most PDE families. We also conclude that our DiffusionPDE becomes insensitive to the number of observations once more than 3% of the points are observed.

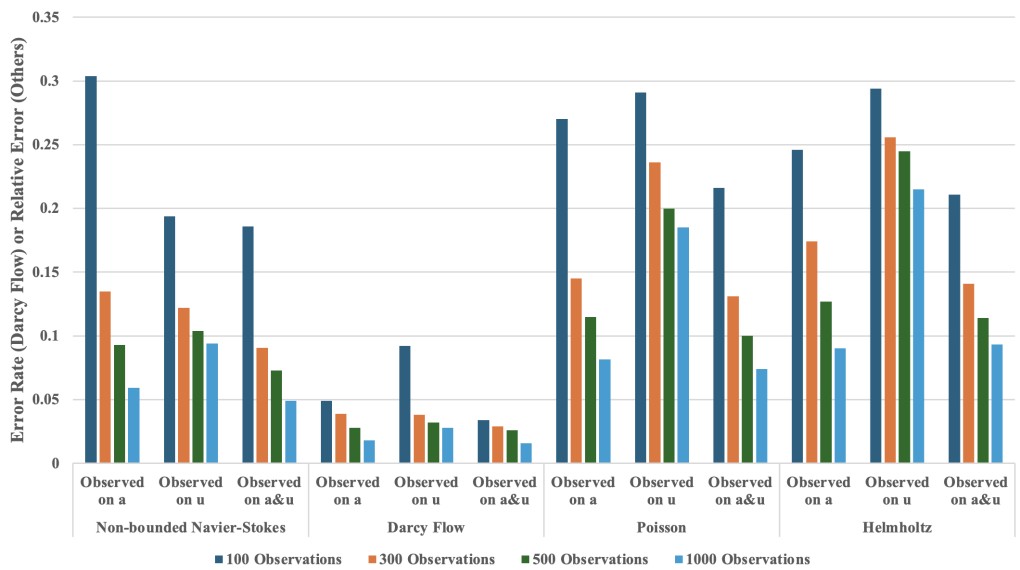

(a) Error rates for Darcy Flow and relative errors for other PDEs of recovered coefficients or initial states $a$.

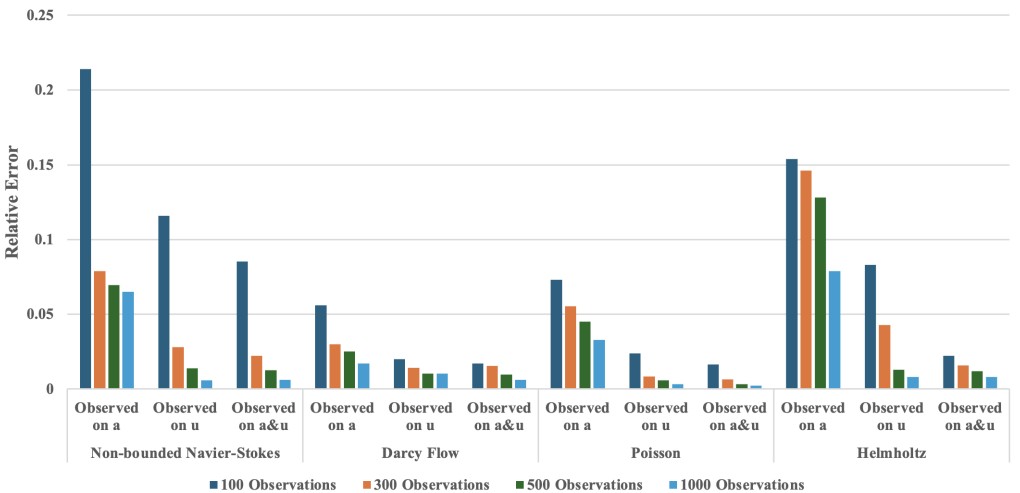

(b) Relative errors of recovered solutions or final states $u$.

Figure 21: Error rate or relative error of both coefficients (or initial states) $a$ and solutions (or final states) $u$ with different numbers of observations.

## M Solving Forward and Inverse Problems across Varied Resolutions

To evaluate the generalizability of DiffusionPDE, we implemented the model on various resolutions, including $64 \times 64$ and $256 \times 256$, while maintaining the same percentage of observed points. For resolutions of $64 \times 64$, $128 \times 128$, and $256 \times 256$, we observe $125$, $500$, and $2000$ points on $a$ or $u$ respectively, which are approximately $3\%$ for each resolution. Overall, DiffusionPDE is capable of handling different resolutions effectively. For instance, Table 9 presents the forward relative errors of the solution $u$ and inverse error rates of the coefficient $a$ for the Darcy Flow, demonstrating that DiffusionPDE performs consistently well with similar error rates across various resolutions.

Table 9: Forward relative errors and inverse error rates of Darcy Flow across different resolutions.

| Resolution | Forward Relative Error | Inverse Error Rate |
|:---:|:---:|:---:|
| $64 \times 64$ | 2.9% | 4.3% |
| $128 \times 128$ | 2.5% | 3.2% |
| $256 \times 256$ | 3.1% | 4.1% |

# N    Comparison with Other Baselines

We have compared the results using the RBF kernel [60], as shown in Fig. 22. For the forward process of solving the Poisson, Helmholtz, and Darcy Flow equations, the RBF kernel achieved solution errors of approximately $14.3\%$, $23.1\%$, and $18.4\%$, respectively, with 500 random observation points. However, when addressing the inverse problem, the errors increased significantly to $141.2\%$, $143.1\%$, and $34.0\%$, respectively. This increase in error is likely due to the inherent challenges of solving inverse problems with such a straightforward method.

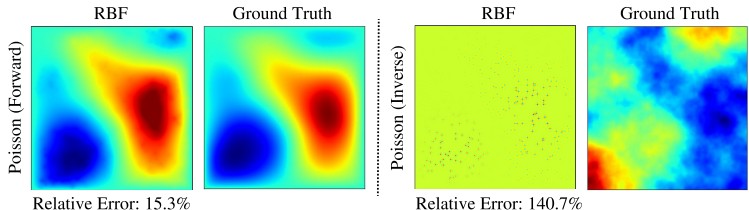

Figure 22: Forward and Inverse Results of Poisson equation recovered by 500 observation points using RBF Kernel.

Additionally, we compare our DiffusionPDE method with a single U-Net model. The U-Net is trained based on our EDM diffusion model, where we initially train it to map between 500 fixed input points and the full output space, as illustrated in Fig. 23. For the Navier-Stokes equation, the prediction of the final state results in an average test error of approximately 39%, which is significantly higher than the error produced by our diffusion model. Furthermore, when making predictions using 500 different sampling points, the relative error increases to approximately 49%. We also train another U-Net model to map between 500 random input points and the full output space, but this model results in a test error of 101%, indicating that the U-Net struggles to adapt to varying sampling patterns and fails to flexibly solve different configurations.

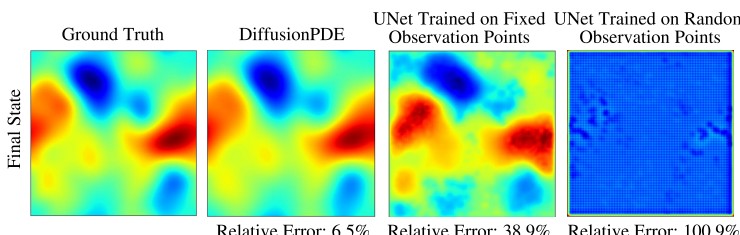

Figure 23: Comparison between DiffusionPDE and U-Net regarding non-bounded Navier-Stokes equation.

