# OpenReview forum: "DiffusionPDE: Generative PDE-Solving under Partial Observation"
_NeurIPS.cc/2024/Conference — NeurIPS 2024 poster_

### Official Review · Reviewer_y14J · 2024-07-08

**Soundness:** 3
**Presentation:** 3
**Contribution:** 2
**Rating:** 5
**Confidence:** 4

**Summary:**

This paper introduces diffusion methods to tackle the partially observed PDEs, named DiffusionPDE. By learning the joint distribution of solution and coefficient space, the proposed model can handle both forward and inverse problems. The authors experiment with diverse PDEs and settings to demonstrate the model's effectiveness.

**Strengths:**

-	This paper successfully utilizes the diffusion methods in solving PDEs, covering both forward and inverse problems.

-	The main text and supplementary materials provide diverse experiment settings, which can well support the model’s effectiveness on partial observations.

-	This paper is overall clear and well-written.

**Weaknesses:**

1.	The technical contribution is limited.

From a technical view, this paper is an application of the diffusion model in PDE solving. There are also some previous methods that also use diffusion methods and leverage the PDE loss [1]. Thus, I think the technical novelty is limited.

[1] A Physics-informed Diffusion Model for High-fidelity Flow Field Reconstruction, JCP 2023

2. Some powerful baselines are missing.

- According to Figure 1, I think the base model of DiffusionPDE is U-Net. How about comparing it with a single U-Net? I think U-Net could be a powerful baseline.

- There are also some latest models that are good at processing partially observed or irregularly placed PDEs, such as OFormer [1] and Transolver [2]. They should include them as baselines.

[1] Transformer for Partial Differential Equations' Operator Learning, TMLR 2023

[2] Transolver: A Fast Transformer Solver for PDEs on General Geometries, ICML 2024

3. Model efficiency comparisons are needed, including GPU memory and running time.

4. I think the proposed model cannot predict the future evolution of a time-dependent PDE. Current tasks are all about “reconstruction” or “imputation”.

**Questions:**

1.	Figure 4, do both forward and inverse tasks use the same diffusion model? Or do we need to train two models for these two different tasks?

2.	I think the base model is U-Net. So how does DiffusionPDE handle the spatially scattered partial observations? Is the input still in the regular grid, but only the sampled locations have ground-truth values?

**Limitations:**

I appreciate that they have discussed the limitations. But I think the mentioned issues about efficiency and limitations on temporal modeling are not trivial. More discussions are expected.

---

> ### Author Rebuttal · Authors · 2024-08-07
>
> Thank you for your thoughtful comments! We are happy that you find our paper provides diverse experiments and is well-written.
>
> > **“The technical contribution is limited.”**
>
> Please refer to the common response above.
>
> > **“Some powerful baselines are missing.”**
>
> 1. U-Net: We trained a U-Net model based on our EDM diffusion model. Initially, we trained the model to map between 500 fixed points in the input space and the full output space (Figure 9  in the PDF). For the Navier-Stokes equation, the prediction of the final state resulted in an average test error of approximately 39.1%, which is significantly higher than the error of our diffusion model. Additionally, we made predictions for the final state using 500 different sampling points, which increased the relative error to approximately 49.2%. Furthermore, we trained another U-Net model to map between 500 random points in the input space and the full output space. This U-Net model resulted in a 101% error during testing, indicating that the model fails to flexibly solve different patterns.
>
> 2. OFormer [1]: We examined the forward and inverse process of Navier-Stokes equation for OFormer taking 500 observation points, and the relative errors are approximately 17% and 23% respectively, which are much larger than DiffusionPDE, as shown in Figure 10 in the PDF file.
>
> 3. Transolver [2]: We believe that Transolver is applied to 3D geometric PDEs and is not suitable for comparison with our DiffusionPDE without significant modifications to their method.
>
> > **“Model efficiency comparisons”**
>
> Please see the common response.
>
> > **“time-dependent PDE”**
>
> Please see the common response.
>
> > **Q1: “Figure 4, do both forward and inverse tasks use the same diffusion model?”**
>
> Yes. We can do forward and inverse tasks with a single model.
>
> > **Q2: “So how does DiffusionPDE handle the spatially scattered partial observations?”**
>
> We use a 128*128 grid to represent the different spaces/states of the PDEs, so yes the input is still in the regular grid. This is a common practice for most works in area [3][4]. In addition, we leverage DPS [5] to handle the spatially scattered partial observations, which is a technique designed for noisy inverse problems (e.g. inpainting).
>
> [1] Transformer for Partial Differential Equations' Operator Learning. Li et. al. TMLR 2023.
>
> [2] Transolver: A Fast Transformer Solver for PDEs on General Geometries. Wu et. al. ICML 2024.
>
> [3] Physics-informed neural networks. Raissi et. al. JCP 2019.
>
> [4] Fourier Neural Operator for Parametric Partial Differential Equations. Li et. al. ICLR 2021.
>
> [5] Diffusion Posterior Sampling for General Noisy Inverse Problems. Chung et. al. ICLR 2023.

---

> > ### Comment · Reviewer_y14J · 2024-08-13
> >
> > I would like to thank the authors' response. After rebuttal, I am clear about the paper's technical contribution.
> >
> > There are still some questions remain. I think the extension of DiffusionPDE to a prediction model is non-trivial. The current framework cannot support this. Also, as the author mentioned, they treat the sparse observation in uniform grids, which is also not a generalizable design since the partial observations can be placed in any position of the domain. Some techniques, such as OFormer, may be more applicable than U-Net.
> >
> > After rebuttal, I think the application scope of DiffusionPDE is limited. The authors should discuss more about the above-mentioned limitations in the revised paper. However, I appreciate the design in the joint modeling of forward-inverse problems. Thus, I improve my score to 5.

---

> ### Author Response · Authors · 2024-08-13
> **Clarification on DiffusionPDE’s Robust Performance Using Continuous Coordinates and Bilinear Interpolation**
>
> We greatly appreciate your response and recognition of our method. We would like to clarify that DiffusionPDE can utilize continuous coordinates with bilinear interpolation in our prediction space to obtain predicted values for points that are not on the grid. By doing so, for example, the forward problem of the non-bounded Navier-Stokes equation results in a relative error of approximately 6.0%, which is comparable to the scenario without interpolation (approximately 6.9%) and even smaller as more points are considered during the calculation of observation loss, while still remaining significantly lower than OFormer (approximately 16.2%). Overall, the diffusion model is powerful, enabling DiffusionPDE to consistently outperform OFormer. We are happy to mention all these discussions in the revised paper.

---

### Official Review · Reviewer_AaET · 2024-07-08

**Soundness:** 3
**Presentation:** 3
**Contribution:** 2
**Rating:** 6
**Confidence:** 4

**Summary:**

The paper proposes to solve PDEs given only sparse measurements by jointly modeling the solution and coefficient space (e.g. the initial conditions) using a diffusion model. By applying diffusion posterior sampling (DPS) the authors obtain samples that are consistent with the sparse measurements and the underlying PDE equations. Several experiments show superior performance of the method compared to standard baselines such as PINNs and FNOs.

**Strengths:**

- Solving PDEs under partial observation is an important problem in real-world applications
- The proposed method is technically sound and improves upon existing baseline methods (PINN, FNO) that do not work well for sparse measurements
- Leveraging a pretrained diffusion model as a generative prior to model the joint distribution of solution and coefficient space is a good idea
- The presentation of the method is clear and supported by concise algorithms and equations. The paper is well written overall
- Experiments consider standard baseline methods for PDEs and cover a sufficient range of different dynamics

----

Post-rebuttal: the authors have addressed quite a few of the initial concerns, and while some concerns (e.g. about the magnitude of the contributions remain), I'd be happy to support an accept. I've raised my score accordingly.

**Weaknesses:**

- The main weakness of the method is the limited novelty. Both sparse measurements and physics-based losses have been considered together with diffusion models, see e.g. Shu et al. (2023). So it seems to me that the main technical novelty is to apply diffusion models to model the joint distribution of two simulation states at different points in time and apply DPS during inference for consistency with the sparse measurements and PDE constraints.
- The experiments do not take into account any stochasticity or uncertainty. In principle, DPS will give a distribution of solutions, which is not the case for the other baseline methods, but this is not explored further in the paper.
- Since the joint distribution models two states at time 0 and time T (for all experiments except Burgers' equation) and $0 \ll T$, the authors need to simplify the PDE loss $\mathcal{L}_{pde}$ to drop any time derivatives. This is a serious limitation.
- It is not clear if DPS works better than classifier-free guidance, as used e.g. in Shu et al. (2023), or other methods for solving inverse problems with diffusion models.
- DPS requires a lot of compute during inference for calculating $\mathcal{L}_{pde}$. For a fair comparison, it would be important to show the number of parameters, training time and inference time for all methods.

**Questions:**

- Algorithm 1 shows an adaption of DPS to EDM (Karras et al. 2022). Is this adaptation novel? Can the authors give some intuition why they apply the DPS losses in line 12 and 13 to the 2nd order correction (line 8) and not apply any trapezoidal rules in this case?
- Are sparse measurements located on a grid that matches the resolution of the diffusion model or do they have continuous coordinates? In the second case, how are they interpolated to match the data resolution of the diffusion model? Does that make classifier-free guidance difficult to apply?
- As noted in the weaknesses: why not use classifier-free guidance? I would like to see a discussion of different methods for inverse problems and diffusion models that can be used here instead of DPS and what are the advantages of using DPS. Reconstructing the solution/coefficient space from sparse measurements alone is a linear inverse problem with a number of different methods that can be used (e.g. Denoising Diffusion Restoration Models; Kawar et al. 2022, among many others) which oftentimes have much nicer theoretical guarantees/higher quality reconstructions and faster sampling speed. When considering these methods, is adding the PDE loss $\mathcal{L}_{pde}$ and thus making the problem a non-linear inverse problem really beneficial?
- Likewise, as mentioned above: what are the parameter counts and runtimes of the method and the baselines?

**Limitations:**

The authors have mentioned slow sampling speed as a limitation in the conclusion, but I think an extended discussion of this would be important to include.

---

> ### Author Rebuttal · Authors · 2024-08-07
>
> Thank you for agreeing with us that we consider an important problem and propose a technically sound method.
>
> > **“The main weakness of the method is the limited novelty”**
>
> Please see our common response above.
>
> > **“The experiments do not take into account any stochasticity or uncertainty”**
>
> On one hand, we use a deterministic diffusion model. Thus, given partial observations as input, the only stochasticity or uncertainty in our method is the initial random noise. On the other hand, we agree it is important to take this into consideration and hence supplement an experiment where we test the effect of different noise seeds on the initial and final state of Navier-Stokes equations. Please see Figure 6 in the attached PDF. We promise to add this to the final version of the paper.
>
> > **“The authors need to… drop any time derivatives”**
>
> Please see the common response.
>
> > **Q1: “Can the authors give some intuition why they apply the DPS losses in lines 12 and 13 to the 2nd order correction (line 8) and not apply any trapezoidal rules in this case?”**
>
> DPS guides the sampling procedure by the end of each iteration, and we are not specifically modifying it.
>
> > **Q2: “Are sparse measurements located on a grid?”**
>
> Yes, we use a 128*128 grid to represent the different spaces/states of the PDEs and the sparse measurements lie on the grid points. This is a common practice for most works in area [1][2].  However, we can also easily extend our method to be not restricted to grid points by interpolating on the grid and supervising the inerpolated value.
>
> > **Q3: “It is not clear if DPS works better than classifier-free guidance”**
>
> In Figure 8 in the PDF, we evaluate the forward and inverse processes of the non-bounded Navier-Stokes equation, comparing DiffusionPDE with Diffusion with CFG when only initial and final states are considered. Our DiffusionPDE method achieves lower relative errors in both evaluations. We also compare our results with those of Shu et al. [3], where full time intervals are solved autoregressively. In this approach, the error of the final state increases to approximately 13%, which is higher than that of the two-state model (see Figure 10 in the PDF file).
>
> > **Q4: “Show the number of parameters, training time and inference time for all methods.”**
>
> Please see the common response.
>
> [1] Physics-informed neural networks. Raissi et. al. JCP 2019.
>
> [2] Fourier Neural Operator for Parametric Partial Differential Equations. Li et. al. ICLR 2021.
>
> [3] A Physics-informed Diffusion Model for High-fidelity Flow Field Reconstruction. Shu et al. Journal of Computational Physics 2023.

---

> > ### Comment · Reviewer_AaET · 2024-08-13
> > **Rebuttal**
> >
> > Thank you for the updates and the interesting new results in the PDF.
> >
> > It's definitely an interesting direction, and I'm open to supporting acceptance by raising my score. I still find it difficult to strongly argue for acceptance given the limited scope of the technical contributions, though.

---

> > > ### Author Response · Authors · 2024-08-13
> > >
> > > We greatly appreciate your response. We would appreciate it if you could let us know of any specific questions we can address to help raise the score.

---

### Official Review · Reviewer_4PC9 · 2024-07-10

**Soundness:** 2
**Presentation:** 3
**Contribution:** 2
**Rating:** 4
**Confidence:** 3

**Summary:**

The work uses a guided diffusion process to solve the PDE forward and inverse problems with partial observations. Instead of learning the parameter-to-solution map ($a\rightarrow u$) as in Neural Operators, the method learns the diffusion process on the joint distribution $(a,u)$, and use guided diffusion for inference under sparse observations. Compared with several baseline method, the proposed method shows improved performance for solving forward and inverse problem with sparse observations.

**Strengths:**

The work uses a guided diffusion process to solve the PDE forward and inverse problems with partial observations.The authors compare with several baseline methods. The idea is clearly presented, and might be useful for the community.

**Weaknesses:**

The paper presents an interesting approach to solving PDE forward or inverse problems with sparse observations, which is an appealing concept given the minimal data requirement. However, this approach raises some concerns about the well-posedness of the problem. For example, in forward problems where sparse observations of the parameter $a(x_i)$ are available, there are infinitely many ways to interpolate $a$ and solve the PDE to obtain $u$. They are all valid solutions that satisfy both the PDE and the observations. This suggests that the method's ability to achieve good recovery might heavily rely on the strong regularization imposed by the training dataset, potentially limiting its practical utility as it may only favor solutions resembling those in the training set.

Additionally, in Appendix C, Table 2, the weightings for observation and PDE loss are significantly higher (by two to six orders of magnitude) than those for $\nabla_x \log(p(x))$ as described in Equation 8, which might indicate a predominance of data fitting over the diffusion process. It would be beneficial if the authors could provide more guidance on how these weights were chosen and discuss the implications of using smaller weights. Understanding the rationale behind these choices could help clarify the model's dependency on these parameters and their impact on the solution's behavior.

**Questions:**

(1) The results from the baseline methods (PINO, DeepONet, PINNs, and FNO) are so bad. The claim that these methods are "not easily extendable" invites further scrutiny:

(a) All the baseline models are supposed to represent smooth functions. However, in Figure 4, 7, 8, they look discontinuous at the training points.
An explanation of how these models were trained and how inference was conducted could clarify why these discrepancies appear.

(b) Taking PINNs as an example in the Darcy flow problem.
Let $\hat{a}(x)$ and $\hat{u}(x)$ be the (potentially noisy) observation at $x$.
We can represent $a(x)$ by a neural network, $a_V(x)$ (use neural net for convenience, could be other representations), and the PDE solution $u(x)$ by a neural net $u_W(x)$, where $V$ and $W$ are the weights of the neural network.
We can solve the following optimization problem:

$$\min_{V,W} \sum_{x\in T_d} (u_W(x) - \hat{u}(x))^2 + (a_V(x) - \hat{a}(x))^2 + \sum_{x \in T_r} (\nabla \cdot (a_V(x) \nabla u_W(x)) - q(x))^2$$

where $T_d$ is the observational point, and $T_r$ is the residual points, which does not need to be the same as $T_d$.

(c) Similar, for a trained neural operator parametrized by W, $G_W[a] = u$. We can solve the following optimization problem:

$$\min_V \sum_{x\in T_d} (a_V(x) - \hat{a}(x))^2 + (G\[a_V\](x)-\hat{u}(x))^2$$

where $G[a_V]$ is the solution of the PDE with parameter $a_V$.

It seems that all the baseline methods can be used for forward and inverse problems with sparse observation.
It is unclear why the proposed method would offer superior performance compared with the baselines.

In contexts where full observation are available, as shown in Table 4, one might intuitively expect methods like PINNs—which utilize residual losses to ensure adherence to PDE constraints—or Neural Operators—which establish a direct parameter-to-solution mapping—to deliver more accurate results compared to a method that relies on a diffusion process. This leads to a critical inquiry on why diffusion process gives better accuracy for PDE problems.

(2) How is the PDE loss computed? Is it by finite difference on a regular grid? Detailing this in the main text could help readers assess the accuracy and applicability of the PDE loss in different scenarios.

**Limitations:**

The author mention several limitation in the conclusion.

---

> ### Author Rebuttal · Authors · 2024-08-07
>
> Thank you for the detailed and constructive review!
>
> > **“The method’s success may heavily depend on the strong regularization from the training dataset.”**
>
> We use the same data generation methods as other studies [1, 2, 3], ensuring fair comparisons. The process does not favor any specific subset of PDEs. It’s reasonable to assume that a dataset can be obtained containing patterns or regularization about field distributions. For example, factories manufacture parts with material properties following specific distributions, and our method can solve PDEs for these parts. While our method may struggle with out-of-distribution cases, we know of no existing method that performs well under sparse observations without assumptions about PDE coefficients to the best of our knowledge. This is an open and exciting field with great potential.
>
> > **“The weightings for observation and PDE loss are significantly higher.”**
>
> The weights depend on the scale of the coefficients or solutions. In our static PDE dataset, the absolute values of $u$ are much smaller than those of $a$. To ensure their distributions fall between -1 and 1, $a$ and $u$ are scaled differently during dataset preparation. During inference, they are scaled back to their original values to calculate PDE and observation losses. Thus, $\zeta_u$ should be larger than $\zeta_a$ to ensure the accurate recovery of both $a$ and $u$, as $L_u$ is much smaller than $L_a$. For example, in the Darcy Flow equation, $a$ is 3 or 12, while the absolute value of $u$ can be less than 0.01, making $L_u$ approximately $10^{-4}$ times $L_a$, leading to $\zeta_u$ being $10^4$ times $\zeta_a$. Similarly, the choice of $\zeta_{pde}$ is also influenced by the data scale.
>
> > **Q1(a): “The baseline models… look discontinuous”**
>
> All models are trained with full observations. During inference, we apply a mask to set non-observed input values to zero, which can cause output discontinuities, especially with FNO as the model does not guarantee smooth output. Results are smoother after applying the optimization from Question 1(b) (Figure 5 in the PDF). We also considered training on partial observations (Appendix G), where the model maps sparse inputs to full outputs, but this method doesn’t generalize to different sampling patterns. Additionally, using a different PINN codebase with modified hyperparameters [4] results in smoother outputs for the Burgers’ equation (Figure 4 in the PDF).
>
> > **Q1(b)(c): Suggest optimizing baselines methods**
>
> We acknowledge that optimizing the sampling space is another way to run the baselines. We applied this approach, and the results are shown in Table C below and Figure 5 in the PDF. Optimization reduces errors and smooths the solutions. However, the resulting values are smaller due to the smoothing effect from minimizing PDE loss, and the overall error compared to the ground truth remains much higher than DiffusionPDE. This may be due to the difficulty in optimizing the derivatives of noisy $a$ and $u$. Additionally, optimizing FNO and PINO is slow. While other models were optimized within 30 seconds, FNO and PINO took over 10 minutes to converge. Table B above shows these optimized baselines are slower than DiffusionPDE, likely due to the computational complexity of the Fourier transform.
>
> **Table C: Relative errors of solutions (or final states) and coefficients (or initial states) when solving forward and inverse problems respectively with sparse observations after optimizing the baselines. Error rates are used for the inverse problem of Darcy Flow.**
> | | | DiffusionPDE | DeepONet | PINO | FNO |  PINNs  |
> |-|-|:-|:-:|:-:|:-:|:-:|
> | **Darcy Flow** | Forward | **2.5%** | 31.3% | 32.6% | 27.8% | 6.9% |
> | | Inverse |**3.2%**|41.1%|49.2%| 49.3% | 59.7%|
> | **Poisson**| Forward | **4.5%** |73.6%| 79.1%  |  70.5% |  77.8%  |
> | | Inverse | **20.0%** | 75.0% | 115.0% | 118.5%  |  73.9%  |
> | **Helmholtz** | Forward | **8.8%** | 77.6% | 67.7%  |  84.8%  |  79.2%  |
> | | Inverse |  **22.6%**   | 100.7%   | 125.3% | 131.6%  | 103.7%  |
> | **Non-bounded NS** | Forward | **6.9%** | 96.5% | 93.3%  |  91.6%  | 106.1%  |
> | |Inverse |  **10.4%**   |  71.9% | 87.8%  |  89.3%  | 108.6%  |
> | **Bounded NS** | Forward | **3.9%** |  89.1% | 80.8%  |  81.2%  |  84.4%  |
> | | Inverse |   **2.7%**   |  88.6%   | 47.3% | 48.7% |  82.1% |
>
> > **Q1(c): “Why diffusion process gives better accuracy for PDE problems than other methods like PINNs”**
>
> For partial observations, PINNs [1] does not learn any knowledge about the PDE, making it difficult to inpaint the missing parts. Neural operators [2] learn to map the entire coefficient space to the entire solution space and hence have difficulty taking partial observations as input. For full observations, PINNs require iterations of optimization to converge to the solution, which can be vulnerable to local minimal or failure to convergence, leading to higher error. It is also less robust compared to our data-driven method. Our method, on the other hand, enjoys the advantage of combining PDE knowledge and observation guidance through an iterative generative model. Iterative generative models, such as diffusion models, tend to beat feedforward models like GANs.
>
> > **Q2: “How is the PDE loss computed?”**
>
> Sobel filters are applied to the grid with asymmetric padding to compute derivatives. The PDE loss at the boundary is manually set to zero to handle edge effects. This procedure is applied to the pixel space rather than the continuous coordinate space. We promise to supplement these details in the final version of the paper.
>
> [1] Physics-informed neural networks. Raissi et. al. JCP 2019.
>
> [2] Fourier Neural Operator for Parametric Partial Differential Equations. Li et. al. ICLR 2021.
>
> [3] Learning to solve pde-constrained inverse problems with graph networks. Zhao et. al. ICML 2022.
>
> [4] Physics-informed deep-learning applications to experimental fluid mechanics. Eivazi et. al. Measurement Science and Technology 2024.

---

> > ### Comment · Reviewer_4PC9 · 2024-08-13
> >
> > Thanks for the the hard work and for conducting additional experiments. I appreciate the comprehensive benchmarking and the detailed responses to the reviewers' comments.
> >
> > > we know of no existing method that performs well under sparse observations without assumptions about PDE coefficients to the best of our knowledge.
> >
> > I agree. But I'm not asking for out-of-sample distribution.
> > My questions is on the well-posedness of the problem.
> >
> > Using the Darcy flow as an example. Suppose $a$ is sampled from some Gaussian random field with certain correlation structure.
> > If we only have partial observation of $\{x_i, a(x_i)\}$, for i = 1,...,N, and N is relatively small. Suppose the ground truth is $a_{GT}$. However, there could be $a_1$, ..., $a_M$ that are all consistent with the observation. That is, $a_j(x_i) = a_{GT}(x_i)$ for i = 1,...,N. and j = 1,...,M.
> >
> > In this case, DiffusePDE might recover $a_{GT}$ (hypothetically), while FNO, DeepOnet, PINN, etc, might recover $a_1$, ..., $a_M$, which will certainly have error with respect to $a_GT$, but they are all consistent with the observation data and the PDE. Therefore, the low accuracy of other baseline methods does not necessarily mean they are worse. They might be different but still valid solutions to the problem.
> >
> > As an even more straightforward approach in this case, we can sample the Gaussian random field conditional on the observation, and solve the PDE numerically, this would give infinite number of valid solutions to the problem.
> >
> > > During inference, we apply a mask to set non-observed input values to zero.
> >
> > Why do we mask the non-observed input values to zero? It seems a more reasonable approach is to exclude those in the loss function.
> >
> > > For partial observations, PINNs [1] does not learn any knowledge about the PDE, making it difficult to inpaint the missing parts
> >
> > My understanding is that PINN, or any method employing PDE loss, should be able to inpaint missing parts in ways that remain consistent with the PDE. As the PDE loss can be computed and minimized without any data.
> >
> > > Neural operators [2] learn to map the entire coefficient space to the entire solution space and hence have difficulty taking partial observations as input
> >
> > This statement appears to conflict with the typical characterization of neural operators (FNO or DeepONet) as being "resolution invariant" -- the neural operator can be evaluated at any point, even those not in the training points.
> > Could you provide more details on the difficulties encountered?
> >
> >
> > > For full observations, PINNs require iterations of optimization to converge to the solution, which can be vulnerable to local minimal or failure to convergence, leading to higher error.
> >
> > This seems to suggest that PINN fail to train in these cases, which invite questions on how is PINN trained, As the problems (Darcy flow, Poisson, etc) are standard examples in PINN literatures [1,2,3,4].
> >
> > > Iterative generative models, such as diffusion models, tend to beat feedforward models like GANs.
> >
> > I'm not concerned about GANs. To my understanding, in your approach, the joint distribution between the parameter $a$ and the solution $u$ is modeled. That is, for any particular $a$, there is a conditional distribution of $u$.
> > However, for PDEs, the solution $u$ should be uniquely determined by the parameter $a$. And neural operators aim to learn this mapping.
> > It seems somewhat counter-intuitive that learning a joint distribution could be more effective than learning this mapping directly.
> >
> > As a summary, due to ongoing concerns regarding the problem formulation, implementation of some of the baseline method, and an lack of understanding of advantages of the diffusion-based approach, I will maintain my current score for this review.
> >
> > [1] M. Raissi, P. Perdikaris, and G. E. Karniadakis, “Physics-informed neural networks: A deep learning framework for solving forward and inverse problems involving nonlinear partial differential equations,”
> >
> > [2] DeepXDE(https://deepxde.readthedocs.io/en/latest/demos/pinn_forward.html)
> >
> > [3] Q. He and A. M. Tartakovsky, “Physics-Informed Neural Network Method for Forward and Backward Advection-Dispersion Equations,”
> >
> > [4] S. Wang, S. Sankaran, H. Wang, and P. Perdikaris, “An Expert’s Guide to Training Physics-informed Neural Networks,”

---

> > > ### Author Response · Authors · 2024-08-13
> > >
> > > We greatly appreciate your response. We would like to clarify a few points:
> > >
> > > > "the well-posedness of the problem"
> > >
> > > We agree that there are infinitely many valid solutions to the partial observations. However, we assume that there are statistical patterns in the coefficient and solution spaces, which means that each valid solution has its own 'probability' of actually being the solution. We aim to learn this probability distribution with our diffusion model, which excels at sampling a highly likely full state given partial observations, as reflected by the much lower error compared to all prior methods suggested by ourselves and other reviewers, including a simple baseline (suggested by reviewer nBsW) of completing the coefficient space using an RBF kernel. Such problems of statistically inferring unobserved data have been widely studied across domains, such as image inpainting in computer vision and matrix completion in machine learning, which are deemed well-posed problems of maximizing probability.
> > >
> > > > "Why do we mask the non-observed input values to zero?"
> > >
> > > For our method and other baselines that require optimization during inference, we indeed exclude non-observed values. For baseline methods that expect a complete input space, e.g., FNO, we have to fill in those values. We have tried filling in the missing values with constant zero values (default) or using RBF kernel or nearest neighbor interpolation, all resulting in lower performance than DiffusionPDE.
> > >
> > > > "As the PDE loss can be computed and minimized without any data. / This seems to suggest that PINN failed to train in these cases..."
> > >
> > > PINN is trained well according to the original paper, and the reviewer is correct in noting that PINN can handle partial observations and automatically complete the missing data. However, it’s generally understood that PINN tends to have a test error between 10% and 30%, even in fully observed situations, because the training loss never perfectly converges to 0. This high error is one of the reasons why there is growing interest in neural operators [1].
> > >
> > > > "This statement appears to conflict with the typical characterization of neural operators (FNO or DeepONet) as being 'resolution invariant'... Could you provide more details on the difficulties encountered?"
> > >
> > > FNO can solve problems across different resolutions, and the reviewer is correct in noting that the output of neural operators can be evaluated at any point. However, we cannot assume that the input to neural operators can be sparse like ours. In fact, neural operators require a complete continuous grid as input, whereas we only have very sparse observation points (approximately 3%), which are highly discrete and do not meet this requirement. As a result, the model struggles to learn higher-frequency features. To the best of our knowledge, neural operators have not been shown to be effective on highly sparse inputs.
> > >
> > > > "It seems somewhat counter-intuitive that learning a joint distribution could be more effective than learning this mapping directly."
> > >
> > > The neural operators that learn the mapping directly indeed perform comparably to DiffusionPDE given "full" observation of $a$. However, when the observation of $a$ is not complete, the solution $u$ is not uniquely determined. Hence, we need to sample from the conditional distribution of $u$, a task in which DiffusionPDE outperforms neural operators and their variants suggested by the reviewers.
> > >
> > > [1] Physics-Informed Neural Operator for Learning Partial Differential Equations. Li et. al. ACM/JMS Journal of Data Science 2024.

---

### Official Review · Reviewer_nBsW · 2024-07-10

**Soundness:** 4
**Presentation:** 4
**Contribution:** 3
**Rating:** 8
**Confidence:** 5

**Summary:**

The paper uses score based generative diffusion models to find the forward and backwards solution of a set of PDEs given partial observations of the solution and/or incomplete knowledge of the coefficients. The method performs well, and outperforms other ML  methods such as FNO, as well as 'standard' FE type methods, for a range of standard test problems. The method reconstructs This is a novel approach, which delivers good performance, with low errors  at a competitive speed. Extensive tests are given, with careful analysis of the results.

**Strengths:**

The use of score based generative methods in this context, where both the solution and the parameter estimates are updated, is novel. The method is clearly effective for the problems considered and should have good applications to real world examples. Extensive tests on a series of standard test problems show that the errors of the method are much lower than other ML based methods such as FNO.

**Weaknesses:**

This paper suffers as do many similar papers from a limited range of examples. It concentrates on the usual examples of PDEs such as NS and Bergers, and in both cases of these it looks at problems with quite moderate viscosity, which are realtively east to solve. This is more or less inevitable for such a short paper as this, especially as comparisons are needed with other method. But I would have liked to have seen more novel examples than the usual ones. This is not really a criticism of this paper, but is something to consider for future work. It would be imporoved by a fairer comparison with other methods which work with incomplete data and measurements. A clear exanple of this being the data assimilation widely used in physical modelling for just this range of problems. These should be descibed somewhere in the introduction and in Section 2. (Although of course these latter methods are slow in comparison.) The method is also limited (see later) to looking at certain slices of the solution.

**Questions:**

1. How does this method compare with a data assimilation approach
2. How easy would it be to extend the method to full time intervals
3. How easy is it to extend the method to higher dimensions
4. Have the authors tried out the method on more challenging PDE examples.
5. Also consider tests on NS and Bergers' eqn with much smaller viscosity.

**Limitations:**

The model as described only looks at slices of solutions of 2D problems. This has been clearly identified by the authors. In this sense it is vrather limited when compared to other ML based approaches, and of course traditional FE based methods. I am pleased that this is recognised and that the authors plan to address this. The DiffusionPDE method will only be truly competitive when this is done, but this paper is a good step in this direction.

---

> ### Author Rebuttal · Authors · 2024-08-07
>
> Thank you for your positive comments on our work! We feel much encouraged that you recognize the novelty of our work.
>
> > **a fairer comparison with other methods that work with incomplete data and measurements**
>
> In addition to GraphPDE, we further compare our method with OFormer [1], Shu et al. [2], and UNet baselines. Please see Figures 7-10 in the PDF.
>
> > **Q1: data assimilation**
>
> We have compared the results using the RBF kernel (Figure 1 in the PDF). For the forward process of solving the Poisson, Helmholtz, and Darcy Flow equations, the RBF kernel achieved solution errors of approximately 14.3%, 23.1%, and 18.4%, respectively, with 500 random observation points. However, when addressing the inverse problem, the errors increased significantly to 141.2%, 143.1%, and 34.0%, respectively. This increase in error is likely due to the inherent challenges of solving inverse problems with such a straightforward method. We are happy to include such discussions in the revision of the paper.
>
> > **Q2: full time intervals**
>
> Please see our reply in the common response above.
>
> > **Q3: higher dimensions**
>
> Yes, we can extend the method to higher dimensions using state-of-the-art diffusion models designed for such cases. For instance, 3D diffusion models (e.g., [3]), especially latent diffusion models, could be appropriate for handling 3D geometric PDEs.
>
> > **Q4: more challenging PDE examples**
>
> To further demonstrate the generalization capability of our model, we conducted additional tests on different data settings for Darcy Flow. In Figure 2 in the PDF, we solve the forward and inverse problems of Darcy Flow with 500 observation points, adjusting the binary values of $a$ to 20 and 16 instead of the original 12 and 3. Our results indicate that DiffusionPDE performs equally well under these varied data settings, showcasing its robustness and adaptability.
>
> > **Q5: NS and Burgers' equation with much smaller viscosity**
>
> Yes, we also test DiffusionPDE on the Burgers’ equation with a viscosity of $1 \times 10^{-3}$ and on the Navier-Stokes equation with a viscosity of $1 \times 10^{-4}$, which are 10 times smaller than the ones in the main paper. For the Burgers’ equation, we are able to recover the full time interval with 5 fixed sensors at a relative error of approximately 6%, which is close to the error of approximately 2~5% in the main paper. For the Navier-Stokes equation, we can solve the forward and inverse problems with relative errors of approximately 7% and 9%, respectively, using 500 observation points. The errors are also close to the ones in the main paper, where the forward and inverse errors of NS are approximately 7% and 10%. Please see Figure 3 in the PDF.
>
> [1] Transformer for Partial Differential Equations' Operator Learning. Li et. al. TMLR 2023.
>
> [2] A Physics-informed Diffusion Model for High-fidelity Flow Field Reconstruction. Shu et. al. JCP 2023.
>
> [3] 3D Neural Field Generation using Triplane Diffusion. Shue et. al. CVPR 2023.

---

### Author Rebuttal · Authors · 2024-08-07

We would like to thank all reviewers for their constructive feedback! We will first clarify common concerns from the reviewers.

> **The method drops time derivatives and cannot solve for full time intervals (Reviewer nBsW, AaET)**

Our method can in theory support time derivatives and solve for full time intervals, as we have demonstrated in the 1D Burger’s equation (Figure 4 in the paper). Due to our choice of the diffusion model,  which enables channel concatenation, we decided to model the joint distribution on the initial state and an arbitrary time state for other time-dependent PDEs such as the Naiver-Stokes equations. However, we think this is not a limitation of our method because:

1. The choice of diffusion model is orthogonal to our proposed method. For example, we can extend our solutions of the Naiver-Stokes equations to full time intervals by replacing the current 2D diffusion model with ones that support higher dimensions, such as 3D or video diffusion models [3,4,5], without changing the algorithm we described in the paper (Lines 12-14, Algorithm 1).

2. Even at its current state,  where the method is applied to solve time-dependent PDE by modeling the joint distribution between two time states for time-dependent PDEs, our method can achieve more accurate results in solving both forward and inverse PDEs. Plus, our method is faster than methods that autoregressively solve time-dependent PDEs [1].

> **“There is limited novelty of this paper compared with [1] (Shu et. al. 2023)” (Reviewer AaET, y14J)**

We would like to reiterate the novelty of this paper in that we propose to use diffusions to model the joint distribution of different spaces/states of PDEs, achieving state-of-the-art performance in solving forward/inverse PDEs under sparse observation. We would like to emphasize our main novelty/contribution as proposing to use joint distribution as a better way to model PDE problems (as stated in L32-34 in the paper and recognized by reviewer AaET). Compared with [1], we point out a few advantages brought by our method:

1. In terms of PDE types, Shu et al. (2023) is designed for fluid flow fields; as a result, they only show demonstration on limited PDE types such as Naiver-Stokes. Our method can work for a broader category of PDEs, such as Darcy Flow and Poisson equation.

2. In terms of diffusion models, we propose to use DPS [2] as a better choice for the problem. DPS is designed for solving inverse problems and thus suits well the task of solving PDEs given partial observation. We demonstrated that our proposed method can achieve lower errors and better robustness against different sparse sampling patterns (see Figure 7,8 in the PDF).

3. In terms of the problem setup, Shu et al. (2023) requires consistent observations across time. Our method, on the other hand, requires observations at only the initial or final state or both for time-dependent PDEs, enjoying more flexibility on the partial observation.

4. In terms of inference time, Shu et al. (2023) is slower because it is an autoregressive method. In comparison, our method using the joint distribution adopts a unified model that can handle both forward and inverse and shows more accurate results (see Figure 10 in the PDF file).

> **“Show the number of parameters, training time, inference time, and GPU memory” (reviewer AaET, y14J)**

As mentioned in Section 4.2 of our main paper, training the DiffusionPDE model takes approximately 4 hours on 3 A40 GPUs. In comparison, training the PINO or FNO model on a single A40 GPU takes approximately 8 hours, while training DeepONet takes around 40 minutes.

We evaluate the computing cost during the inference stage by testing a single data point on a single A40 GPU for the Navier-Stokes equation, as shown in Table A. DiffusionPDE has a lower computing cost compared to Shu et al. (2023), which autoregressively solves the full time interval. This advantage becomes more significant when we increase the number of time steps.

**Table A: Inference computing cost of sparse-observation-based methods.**
| Method        | DiffusionPDE (Ours) | GraphPDE | Shu et al. (2023) | OFormer |
|---------------|:------------:|:--------:|:-------------:|:-------:|
| **#Parameter (M)**          |   54  | 1.3  | 63 | 1.6  |
| **Inference time (s)**           |   140     |  84  |     180     |   3.2   |
| **GPU memory (GB)**	| 6.8 |  3.6   | 7.2 | 0.1  |

Further, we evaluate the inference runtimes on one single A40 GPU of vanilla full-observation-based methods and also the optimization time of them during the inference as suggested by Reviewer 4PC9. The optimization runtimes are significantly slower, especially when using Fourier transforms.

**Table B: Average inference runtimes (in seconds) of full-observation-based methods with and without optimization.**
| Method          |  PINO  |  FNO   | DeepONet | PINNs  |
|-----------------|:------:|:------:|:--------:|:------:|
| Vanilla |  1.0e0  | 9.8e-1 |  7.4e-1  |  1.5e0 |
| With Optimization | 6.7e2  | 6.7e2  |  3.5e1   |  3.7e1 |

[1] A Physics-informed Diffusion Model for High-fidelity Flow Field Reconstruction. Shu et. al. JCP 2023.

[2] Diffusion Posterior Sampling for General Noisy Inverse Problems. Chung et. al. ICLR 2023.

[3] Diffusion Probabilistic Models for 3D Point Cloud Generation. Luo et al. CVPR 2021.

[4] Flexible Diffusion Modeling of Long Videos. Harvey et al. NeurIPS 2022.

[5] Open-Sora: Democratizing Efficient Video Production for All. Zheng et al. Github.

---

### Decision · Program_Chairs · 2024-09-25

**Decision:**

Accept (poster)

**Comment:**

This diffusion models proposed here solve PDEs with unknown coefficients by modeling the joint distribution of the coefficients and solution space.  An advantage of this generative approach over some other learning-based PDE solvers is that it can flexibly handle partial observations with irregular patterns.  While one borderline-negative review is skeptical of this probabilistic approach, overall we found the authors' rebuttal to these concerns to be convincing.  There were some concerns about the range of experiments and baseline methods compared to in the original manuscript, but a thorough author rebuttal addressed most of these points.  Please be sure that your final manuscript incorporates these additional experiments and comparisons.